# WARFARE: BREAKING THE WATERMARK PROTECTION OF AI-GENERATED CONTENT

## ABSTRACT

AI-Generated Content (AIGC) is gaining great popularity, with many emerging commercial services and applications. These services leverage advanced generative models, such as latent diffusion models and large language models, to generate creative content (e.g., realistic images and fluent sentences) for users. The usage of such generated content needs to be highly regulated, as the service providers need to ensure the users do not violate the usage policies (e.g., abuse for commercialization, generating and distributing unsafe content). A promising solution to achieve this goal is watermarking, which adds unique and imperceptible watermarks on the content for service verification and attribution. Numerous watermarking approaches have been proposed recently. However, in this paper, we show that an adversary can easily break these watermarking mechanisms. Specifically, we consider two possible attacks. (1) Watermark removal: the adversary can easily erase the embedded watermark from the generated content and then use it freely bypassing the regulation of the service provider. (2) Watermark forging: the adversary can create illegal content with forged watermarks from another user, causing the service provider to make wrong attributions. We propose `Warfare`, a unified methodology to achieve both attacks in a holistic way. The key idea is to leverage a pre-trained diffusion model for content processing and a generative adversarial network for watermark removal or forging. We evaluate `Warfare` on different datasets and embedding setups. The results prove that it can achieve high success rates while maintaining the quality of the generated content. Compared to the inference process of existing diffusion model-based attacks, `Warfare` is **5,050∼11,000×** faster.

## 1 INTRODUCTION

Benefiting from the advance of generative deep learning models (Rombach et al., 2022; Touvron et al., 2023), AI-Generated Content (AIGC) has become increasingly prominent. Many commercial services have been released, which leverage large models (e.g., ChatGPT (cha), Midjourney (Mid)) to generate creative content based on users' demands. The rise of AIGC also leads to some legal considerations, and the service provider needs to set up some policies to regulate the usage of generated content. *First*, the generated content is one important intellectual property of the service provider. Many services do not allow users to make it into commercial use (Touvron et al., 2023; Mid). Selling the generated content for financial profit (Sel) will violate this policy and cause legal issues. *Second*, generative models have the potential of outputting unsafe content (Wei et al., 2023; Qi et al., 2023; Liu et al., 2023a; Le et al., 2023), such as fake news (Guo et al., 2021), malicious AI-powered images (Salman et al., 2023; Le et al., 2023), phishing campaigns (Hazell, 2023), and cyberattack payloads (Charan et al., 2023). New laws are established to regulate the generation and distribution of content from deep learning models on the Internet (Gov; Sin; Gui).

As protecting and regulating AIGC become urgent, Google hosted a workshop in June 2023 to discuss the possible solutions against malicious usage of generative models (Barrett et al., 2023). Not surprisingly, the *watermarking* technology is mentioned as a promising defense. By adding invisible specific watermark messages to the generated content (Fernandez et al., 2023; Kirchenbauer et al., 2023; Liu et al., 2023b), the service provider is able to identify the misuse of AIGC and track the corresponding users. A variety of robust watermarking methodologies have been designed, which can be classified into two categories. (1) A general strategy is to make the generative model learn a specific data distribution, which can be decoded by another deep learning model to obtain a secret

message as the watermark (Fernandez et al., 2023; Liu et al., 2023b; Zhao et al., 2023b). (2) The service provider can concatenate a watermark embedding model (Zhu et al., 2018; Tancik et al., 2020) after the generative model to make the final output contain watermarks. A very recent work from DeepMind, SynthID Beta (Syn), detects AI-generated images by adding watermarks to the generated images[1]. According to its description, this service possibly follows a similar strategy as StegaStamp (Tancik et al., 2020), which adopts an encoder to embed watermarks into images and a decoder to identify the embedded watermarks in the given images.

The Google workshop (Barrett et al., 2023) reached the consensus that "existing watermarking algorithms only withstand attacks when the adversary has no access to the detection algorithm", and embedding a watermark to a clean image or text "seems harder for the attacker, especially if the watermarking process involves a secret key". However, in this paper, we argue that it is not the case. We find that it is easy for an adversary without any prior knowledge to **remove** or **forge** the embedded secret watermark in AIGC, which will break the IP protection and content regulation. Specifically, (1) a watermark removal attack makes the service providers fail to detect the watermarks which are embedded into the AIGC previously, so the malicious user can circumvent the policy regulation and abuse the content for any purpose. (2) A watermark forging attack can intentionally embed the watermark of a different user into the unsafe content without the knowledge of the secret key. This could lead to wrong attributions and frame up that benign user.

Researchers have proposed several methods to achieve watermark removal attacks (Ulyanov et al., 2018; Liang et al., 2021; Li, 2023; Zhao et al., 2023a; Nam et al., 2021; Wang et al., 2022). However, they suffer from several limitations. For instance, some attacks require the knowledge of clean data (Ulyanov et al., 2018; Liang et al., 2021) or details of watermarking schemes (Nam et al., 2021; Wang et al., 2022), which are not realistic in practice. Some attacks take extremely long time to remove the watermark from one image (Li, 2023; Zhao et al., 2023a). Besides, there are currently no studies towards watermark forging attacks. More detailed analysis can be found in Section 2.2.

To remedy the above issues, we introduce `Warfare`, a novel and efficient methodology to achieve both w̲ater̲mark f̲orge and r̲emoval attacks against AIGC in a unified manner. The key idea is to leverage a pre-trained diffusion model and train a generative adversarial network (GAN) for erasing or embedding watermarks to AIGC. Specifically, the adversary only needs to collect the watermarked AIGC from the target service or a specific user, without any clean content. Then he can adopt a public diffusion model, such as DDPM (Ho et al., 2020), to denoise the collected data. The preprocessing operation of the diffusion model can make the embedded message unrecoverable from the denoised data. Finally, the adversary trains a GAN model to map the data distribution from collected data to denoised data (for watermark removal) or from denoised data to collected data (for watermark forge). After this model is trained, the adversary can adopt the generator to remove or forge the specific watermark for AIGC.

We evaluate our proposed `Warfare` on various datasets (e.g., CIFAR-10, CelebA), and settings (e.g., different watermark lengths, few-shot learning), to show its generalizability. Our results prove that the adversary can successfully remove or forge a specific watermark in the AIGC and keep the content indistinguishable from the original one. This provides concrete evidence that existing watermarking schemes are not reliable, and the community needs to explore more robust watermarking methods. Overall, our contribution can be summarized:

- To the best of our knowledge, it is the **first work** focusing on **removing and forging watermarks** in AIGC under a black-box threat model. `Warfare` is a unified methodology, which can holistically achieve both attack goals. Our study discloses the unreliability and fragility of existing watermarking schemes.

- Different from prior attacks, `Warfare` **does not require the adversary to have clean data or any information about the watermarking schemes**, which is more practical in real-world applications.

- Comprehensive evaluation proves that `Warfare` can remove or forge the watermarks without harming the data quality. It is **time-efficient**, which is 5,050∼11,000× faster than diffusion model attacks during the inference. The total time cost is analyzed in Appendix D.

---

[1]Up to the date of writing, SynthID Beta is still a beta product only provided to a small group of users. Since we do not have access to it, we do not include evaluation results with respect to it in our experiments.

Figure 1: Overview of `Warfare`. (1) Collecting watermarked data from the target AIGC service or Internet. (2) Using a public pre-trained denoising model to purify the watermarked data. (3) Adopting the watermarked and mediator data to train a GAN, which can be used to remove or forge the watermark. $x'$ is the watermarked image. $\hat{x}$ is the mediator image. The subscript $i$ is omitted.

- `Warfare` is effective in the few-shot setting, i.e., it can be **freely adapted to unseen watermarks and out-of-distribution images**. It remains effective for different watermark lengths.

## 2 RELATED WORKS

### 2.1 CONTENT WATERMARK

Driven by the rapid development of large and multi-modal models, there is a renewed interest in generative models, such as ChatGPT (cha) and Stable Diffusion (Rombach et al., 2022), due to their capability of creating high-quality images (Ho et al., 2020; Rombach et al., 2022), texts (cha; Touvron et al., 2023), audios (Kong et al., 2021), and videos (Ho et al., 2022). Such AI-Generated Content (AIGC) can have high IP values and sensitive information. Therefore, it is important to protect and regulate it during its distribution on public platforms, e.g., Twitter (Twi) and Instagram (Ins).

A typical strategy to achieve the above goal is watermarking: the service provider adds a secret and unique message to the content, which can be subsequently extracted for ownership verification and attribution. Existing watermarking schemes can be divided into post hoc methods and prior methods. Post hoc methods convert the clean content into watermarked content using one of the following two strategies. (i) Visible watermark strategy: the service provider adds characters or paintings into the clean content (Liu et al., 2021; Cheng et al., 2018; Wen et al., 2023), which can be recognized by humans. (ii) Invisible watermark strategy: the service provider embeds a specific bit string into the clean content by a pre-trained steganography model (Zhu et al., 2018; Tancik et al., 2020) or signal transformation (Nam et al., 2021), which will be decoded by a verification algorithm later. The steganography approach in our paper specifically stands for methods requiring deep learning models. These methods use a deep learning encoder to embed a secret message in an image. Then a deep learning decoder can extract the message from the image. The signal transformation approach in our paper stands for methods using spread spectrum (SS), improved spread spectrum (ISS), quantization (QT) and so on, to embed message. For different transformations, there exists a specific extraction approach. For prior methods, the generative model directly learns a distribution of watermarked content, which can be decoded by a verification algorithm (Fei et al., 2022; Fernandez et al., 2023; Cui et al., 2023; Zhao et al., 2023b). Specifically, Fei et al. (Fei et al., 2022) designed a watermarking scheme for generative adversarial networks (GANs), by learning the distribution of watermarked images supervised by the watermark decoder. A watermarking scheme (Fernandez et al., 2023; Zhao et al., 2023b) is designed for diffusion models (Rombach et al., 2022), which embeds a predefined bit string into the generated images, and later uses a secret decoder to extract it. The service provider can recognize the AIGC from his generative model or determine the specific user account.

### 2.2 WATERMARK ATTACKS

To the best of our knowledge, one only work (Wang et al., 2021) considers the watermark forging attack. However, they assume the adversary knows the watermarking schemes, which is unrealistic. And they only evaluate LSB- and DCT-based watermarks instead of advanced deep-learning schemes. Other prior works mainly focus on the watermark removal attack. These attack solutions can be summarized into three main categories, i.e., image inpainting methods (Ulyanov et al., 2018; Liang et al., 2021) for visible watermarks, denoising methods (Li, 2023; Zhao et al., 2023a), and disrupting methods (Nam et al., 2021; Wang et al., 2022) for invisible watermarks. However, they have several critical drawbacks in practice. Specifically, the image inpainting methods (Ulyanov et al., 2018; Liang et al., 2021) require clean images and watermarked images to train the inpainting model, which is not feasible in the real world, because the user can only obtain watermarked images from the service providers (Mid). Disrupting methods (Nam et al., 2021; Wang et al., 2022) require the user to know the details of the watermarking schemes, which is also difficult to achieve. The most promising method is based on denoising models. For instance, (Li, 2023) adopted guided diffusion models to

purify the watermarked images and minimize the differences between the watermarked images and diffusion model's outputs. However, using diffusion models to remove the watermark will cost a lot of time. Our `Warfare` aims to address all of these limitations under a black-box threat model.

## 3 PRELIMINARY

### 3.1 SCOPE

In this paper, we target both post hoc and prior watermarking methods. For post hoc methods, we do not consider visible watermarks as they can significantly decrease the visual quality of AIGC, making them less popular for practical adoption. For instance, the Tree-Ring watermark (Wen et al., 2023) is proven to significantly change both pixel and latent spaces (Zhao et al., 2023a), which is treated as "a visible watermark" by Zhao et al. (Zhao et al., 2023a). Hence, it is beyond the scope of this paper. For invisible watermarks, we only consider the steganography approach, as it is much more robust and harder to attack than the signal transformation approach (Nam et al., 2021; Wang et al., 2022; Zhao et al., 2023a). We mainly consider watermarks embedded in the generated images. Watermarks in other other domains, e.g., language, audio, will be our future work.

### 3.2 WATERMARK VERIFICATION SCHEME

We consider the most popular type of secret message used in watermarking implementations: bit strings (Fei et al., 2022; Fernandez et al., 2023; Cui et al., 2023; Zhao et al., 2023b). When a service provider $P$ employs a generative model $\mathcal{M}_G$ to generate creative images for public users, $P$ employs a watermarking scheme (e.g., (Fernandez et al., 2023; Liu et al., 2023b)) to embed a secret user-specific bit string $m$ of length $L$ in each generated image. To verify whether a suspicious image $x^s$ is watermarked by $P$ for a specific user, $P$ uses a pre-trained decoder $\mathcal{M}_D$ to extract the bit string $m^s$ from $x^s$. Then, $P$ calculates the Hamming Distance between $m$ and $m^s$: $\text{HD}(m, m^s)$. If $\text{HD}(m, m^s) \leq (1 - \tau)L$, where $\tau$ is a pre-defined threshold, $P$ will believe that $x^s$ contains the secret watermark $m$.

### 3.3 THREAT MODEL

**Attack Goals.** A malicious user can break this watermarking scheme with two distinct goals. (1) *Watermark removal attack*: the adversary receives a generated image from the service provider, which contains the secret watermark associated with him. He aims to erase the watermark from the generated image, and then use it freely without the constraint of the service policy, as the provider is not able to identify the watermarks and track him anymore. (2) *Watermark forging attack*: the adversary tries to frame up a victim user by forging the victim's watermark on a malicious image (from another model or created by humans). Then the adversary can distribute the image on the Internet. The service provider will attribute to the wrong user.

**Adversary's capability.** We consider the black-box scenario, where the adversary can only obtain the generated image and has no knowledge of the employed generative model or watermark scheme. This is practical, as many service providers only release APIs for users to use their models without leaking any information about the details of the backend models $\mathcal{M}_G$ and $\mathcal{M}_D$. We further assume that all the generated images from the target service are watermark-protected, so the adversary cannot collect any clean images. These assumptions increase the attack difficulty compared to prior works (Ulyanov et al., 2018; Liang et al., 2021; Nam et al., 2021; Wang et al., 2022).

## 4 WARFARE: A UNIFIED ATTACK METHODOLOGY

We introduce `Warfare` to manipulate watermarks with the above goals. Let $x_i$ denote a clean image, and $x_i'$ denote the corresponding watermarked image. These two images are visually indistinguishable. Our goal is to establish a bi-directional mapping $x_i \longleftrightarrow x_i'$. For the watermark removal attack, we can derive $x_i$ from $x_i'$. For the watermark forging attack, we can construct $x_i'$ from $x_i$.

However, it is challenging for the adversary to identify the relationship between $x_i$ and $x_i'$, as he has no access to the clean image $x_i$. To address this issue, the adversary can adopt a pre-trained denoising model to convert $x_i'$ into a mediator image $\hat{x}_i$. Due to the denoising operation, $\hat{x}_i$ is visually different from $x_i$, but does not contain the watermark. It will follow a similar "non-watermarked" distribution as $x_i$. Then the adversary can train a GAN model between $x_i$ and $x_i'$, which is guided by $\hat{x}_i$. Figure 1 shows the overview of `Warfare`, consisting of three steps. Below, we describe the details.

| # of Samples (bit length = 8bit) | Original | | | | | Watermark Remove | | | | | Watermark Forge | | | | |
|---|---|---|---|---|---|---|---|---|---|---|---|---|---|---|---|
| | Bit Acc | FID | PSNR | SSIM | CLIP | Bit Acc↓ | FID↓ | PSNR↑ | SSIM↑ | CLIP↑ | Bit Acc↑ | FID↓ | PSNR↑ | SSIM↑ | CLIP↑ |
| 5000 | | | | | | 49.42% | 20.75 | 24.64 | 0.83 | 0.92 | 96.11% | 18.86 | 24.36 | 0.83 | 0.93 |
| 10000 | | | | | | 50.68% | 23.76 | 24.31 | 0.82 | 0.90 | 98.63% | 15.68 | 24.70 | 0.81 | 0.94 |
| 15000 | 100.00% | 6.19 | 25.23 | 0.83 | 0.99 | 59.88% | 20.32 | 22.87 | 0.80 | 0.92 | 97.80% | 25.34 | 24.55 | 0.80 | 0.92 |
| 20000 | | | | | | 54.59% | 22.90 | 24.93 | 0.84 | 0.90 | 95.99% | 23.56 | 23.74 | 0.80 | 0.92 |
| 25000 | | | | | | 47.80% | 18.42 | 23.59 | 0.83 | 0.91 | 97.84% | 21.09 | 24.94 | 0.82 | 0.93 |

Table 1: Performance of `Warfare` under the different number of collected images on CIFAR-10. The length of embedded bits is 8.

| Bit Length | Original | | | | | Watermark Remove | | | | | Watermark Forge | | | | |
|---|---|---|---|---|---|---|---|---|---|---|---|---|---|---|---|
| | Bit Acc | FID | PSNR | SSIM | CLIP | Bit Acc↓ | FID↓ | PSNR↑ | SSIM↑ | CLIP↑ | Bit Acc↑ | FID↓ | PSNR↑ | SSIM↑ | CLIP↑ |
| 4 bit | 100.00% | 4.22 | 27.81 | 0.89 | 0.99 | 52.53% | 16.36 | 24.51 | 0.86 | 0.92 | 95.76% | 17.59 | 26.70 | 0.88 | 0.94 |
| 8 bit | 100.00% | 6.19 | 25.23 | 0.83 | 0.99 | 47.80% | 18.42 | 23.59 | 0.83 | 0.91 | 97.84% | 21.09 | 24.94 | 0.82 | 0.93 |
| 16 bit | 100.00% | 11.34 | 22.71 | 0.73 | 0.98 | 50.10% | 24.63 | 23.44 | 0.77 | 0.91 | 92.23% | 18.34 | 25.84 | 0.83 | 0.94 |
| 32 bit | 99.99% | 28.76 | 19.99 | 0.53 | 0.96 | 53.64% | 25.33 | 21.17 | 0.64 | 0.91 | 90.14% | 31.13 | 23.41 | 0.71 | 0.93 |

Table 2: Performance of `Warfare` under different bit lengths on CIFAR-10. The number of images for the adversary is 25,000. ↓ means lower is better. ↑ means higher is better.

## 4.1 DATA COLLECTION

The adversary collects a set of images $x_i'$ generated by the target service provider for one user. All the collected data contain one specific watermark $m$ associated with this user. For the watermark removal attack, the adversary can query the service to collect the watermarked images with his own account, from which he aims to remove the watermark. For the watermark forging attack, the adversary can possibly collect such data from the victim user's social account. This is feasible as people enjoy sharing their created content on the Internet and adding tags to indicate the used service[2]. Then the adversary can forge the watermark of the victim user on any images to cause wrong attribution. In either case, a dataset $\mathcal{X}' = \{x_i' | x_i' \sim (\mathcal{M}_G, m)\}$ is established, where $\mathcal{M}_G$ is the service provider's generative model.

## 4.2 DATA PRE-PROCESSING

Given the collected watermarked dataset $\mathcal{X}'$, since the adversary does not have the corresponding non-watermarked dataset $\mathcal{X}$, he cannot directly build the mapping. Instead, he can adopt a public pre-trained denoising model $\mathcal{H}$ to preprocess $\mathcal{X}'$ and obtain the corresponding mediator dataset $\hat{\mathcal{X}}$. The goal of the denoising model is to remove the watermark $m$ from $\mathcal{X}'$. Since existing watermarking schemes are designed to be very robust, we have to increase the denoising strength significantly, in order to distort the embedded watermark. Therefore, we first add very large-scale noise $\epsilon_i$ into $x_i'$ and then apply a diffusion model $\mathcal{H}$ to denoise the images, i.e., $\hat{\mathcal{X}} = \{\hat{x}_i | \mathcal{H}(x_i' + \epsilon_i) = \hat{x}_i, x_i' \in \mathcal{X}', \epsilon_i \in \mathcal{N}(\mathbf{0}, \mathbf{I})\}$. This will make $\hat{x}_i$ highly visually different from $x_i'$ and $x_i$. Figure 6 shows some visualization results of $x_i'$ and $\hat{x}_i$, and we can observe that they keep some similar semantic information but look very different. Table 3 proves that $\hat{x}_i$ does not contain any watermark information due to the injected large noise and strong denoising operation.

The mediator dataset $\hat{\mathcal{X}}$ can be seen as being drawn from the same "non-watermarked" distribution as $\mathcal{X}$, which is different from $\mathcal{X}'$ of the "watermarked" distribution. Therefore, it can help discriminate watermarking images from non-watermarked images and build connections between them. This is achieved in the next step, as detailed below.

## 4.3 MODEL TRAINING

With the watermarked data $x'$ and non-watermarked data $\hat{x}$, the adversary can train a GAN model to add or remove watermarks. This GAN model consists of a generator $\mathcal{G}$ and a discriminator $\mathcal{D}$: $\mathcal{G}$ is used to generate $x$ from $x'$ (watermark removal) or generate $x'$ from $x$ (watermark forging); $\mathcal{D}$ is used to discriminate whether the input is drawn from the distribution of watermarked images $x'$ or the distribution of non-watermarked images $\hat{x}$. Below, we describe these two attacks.

**Watermark removal attack**. In this attack, the generator $\mathcal{G}$ is built to obtain $x$ from $x'$, i.e., $x = \mathcal{G}(x')$, where $x'$ and $x$ should be visually indistinguishable. $x$ generated by $\mathcal{G}$ should make $\mathcal{D}$ believe it is from the same non-watermarked image distribution as $\hat{x}$, because $x$ should be a non-watermarked image. Meanwhile, $\mathcal{D}$ should recognize $x$ as a watermarked image, since it is very close to $x'$. Therefore, the loss functions $L_{\mathcal{G}}$ for $\mathcal{G}$ and $L_{\mathcal{D}}$ for $\mathcal{D}$ are:

$$L_{\mathcal{D}} = -\mathbb{E}_{\hat{x} \in \hat{\mathcal{X}}} \mathcal{D}(\hat{x}) + \mathbb{E}_{x' \in \mathcal{X}'} \mathcal{D}(\mathcal{G}(x')) + w_{\mathcal{D}} \mathbb{E}_{\hat{x} \in \hat{\mathcal{X}}, x' \in \mathcal{X}'} \nabla_{\alpha x' + (1-\alpha)\hat{x}} \mathcal{D}(\alpha x' + (1-\alpha)\hat{x}),$$

$$L_{\mathcal{G}_x} = \mathbb{E}_{x' \in \mathcal{X}'}[\mathrm{L}_1(\mathcal{G}(x'), x') + \mathrm{MSE}(\mathcal{G}(x'), x') + \mathrm{LPIPS}(\mathcal{G}(x'), x')],$$

$$L_{\mathcal{G}_D} = -w_{\mathcal{G}} \mathbb{E}_{x' \in \mathcal{X}'} \mathcal{D}(\mathcal{G}(x')), \quad L_{\mathcal{G}} = L_{\mathcal{G}_D} + w_x L_{\mathcal{G}_x},$$

---

[2]The adversary can collect watermarked content with his own account as well because our method shows strong few-shot power, which can be found in our experiments. The adversary can adopt very few samples to fit an unseen watermark.

| Methods | Original | | | | | Watermark Remove | | | | | Watermark Forge | | | | |
|---|---|---|---|---|---|---|---|---|---|---|---|---|---|---|---|
| | Bit Acc | FID | PSNR | SSIM | CLIP | Bit Acc | FID | PSNR | SSIM | CLIP | Bit Acc | FID | PSNR | SSIM | CLIP |
| CenterCrop | | | | | | 59.89% | - | - | - | 0.90 | 48.33% | - | - | - | 0.93 |
| GaussianNoise | | | | | | 99.92% | 53.80 | 24.97 | 0.71 | 0.86 | 52.28% | 47.07 | 28.64 | 0.75 | 0.89 |
| GaussianBlur | | | | | | 100.00% | 25.09 | 26.26 | 0.84 | 0.86 | 52.10% | 21.18 | 28.17 | 0.88 | 0.89 |
| JPEG | | | | | | 99.27% | 17.42 | 28.40 | 0.89 | 0.89 | 52.19% | 9.96 | 33.36 | 0.94 | 0.90 |
| Brightness | | | | | | 100.00% | 4.26 | 19.70 | 0.87 | 0.95 | 52.28% | 0.39 | 21.16 | 0.91 | 0.98 |
| Gamma | 100.00% | 4.25 | 30.7 | 0.94 | 0.96 | 100.00% | 4.43 | 22.93 | 0.88 | 0.96 | 52.32% | 0.26 | 25.71 | 0.93 | 0.99 |
| Hue | | | | | | 99.99% | 5.93 | 26.84 | 0.93 | 0.94 | 52.21% | 1.60 | 32.06 | 0.98 | 0.97 |
| Contrast | | | | | | 100.00% | 4.26 | 24.28 | 0.85 | 0.95 | 52.33% | 0.25 | 27.62 | 0.90 | 0.98 |
| $DM_s$ | | | | | | 67.82% | 73.30 | 20.61 | 0.62 | 0.69 | 48.78% | 68.91 | 20.89 | 0.64 | 0.70 |
| $DM_l$ | | | | | | **47.20%** | 82.38 | 15.76 | 0.34 | 0.67 | 45.96% | 79.06 | 15.81 | 0.34 | 0.68 |
| $VAE_{SD}$ | | | | | | 65.32% | 43.21 | 19.57 | 0.66 | 0.76 | 49.36% | 40.50 | 19.84 | 0.68 | 0.77 |
| $VAE_C$ | | | | | | 54.36% | 115.79 | 17.42 | 0.43 | 0.72 | 53.90% | 115.19 | 17.47 | 0.43 | 0.72 |
| Warfare | | | | | | 51.98% | 9.93 | 26.61 | 0.91 | 0.90 | **99.11%** | 8.75 | 24.92 | 0.90 | 0.92 |

Table 3: Results of different attacks on CelebA. The bit string length is 32 bits. Best results in **Bold**. Second best results with Underline.

| Methods | Original | | | | | Watermark Remove | | | | | Watermark Forge | | | | |
|---|---|---|---|---|---|---|---|---|---|---|---|---|---|---|---|
| | Bit Acc | FID | PSNR | SSIM | CLIP | Bit Acc | FID | PSNR | SSIM | CLIP | Bit Acc | FID | PSNR | SSIM | CLIP |
| $DM_s$ | | | | | | 71.54% | 78.67 | 20.21 | 0.60 | 0.69 | 49.35% | 69.09 | 20.92 | 0.64 | 0.71 |
| $DM_l$ | | | | | | 53.75% | 82.94 | 15.67 | 0.33 | 0.67 | 50.99% | 81.66 | 15.82 | 0.34 | 0.68 |
| $VAE_{SD}$ | 100.00% | 13.59 | 27.13 | 0.90 | 0.93 | 67.38% | 50.35 | 19.11 | 0.64 | 0.74 | 50.60% | 40.50 | 19.84 | 0.68 | 0.77 |
| $VAE_C$ | | | | | | **49.90%** | 116.75 | 17.35 | 0.42 | 0.71 | 49.09% | 115.19 | 17.47 | 0.43 | 0.72 |
| Warfare | | | | | | 54.36% | 19.98 | 25.29 | 0.88 | 0.88 | **94.61%** | 12.14 | 23.04 | 0.87 | 0.90 |

Table 4: Results of different attacks on CelebA. The bit string length is 48 bits.

where $w_{\mathcal{D}}$, $w_{\mathcal{G}}$, and $w_x$ are the weights for losses and $\alpha$ is a random variable between 0 and 1 (Arjovsky et al., 2017)[3]. $L_1$ is the $L_1$-norm, MSE is the mean squared error loss, and LPIPS is the perceptual loss (Zhang et al., 2018). They can guarantee the quality of the generated image $x$.

**Watermark forging attack**. In this attack, the generator $\mathcal{G}$ is built to obtain $\hat{x}'$ from $\hat{x}$, i.e., $\hat{x}' = \mathcal{G}(\hat{x})$, where $\hat{x}'$ and $\hat{x}$ should be visually indistinguishable. $\hat{x}'$ is the watermarked version of $\hat{x}$. $\hat{x}'$ generated by $\mathcal{G}$ should make $\mathcal{D}$ believe it is from the same watermarked image distribution as $x'$, because $\hat{x}'$ should be a watermarked image. But $\mathcal{D}$ should recognize $\hat{x}'$ as a non-watermarked image, since it is very close to $\hat{x}$. The loss functions $L_{\mathcal{G}}$ for $\mathcal{G}$ and $L_{\mathcal{D}}$ for $\mathcal{D}$ are:

$$L_{\mathcal{D}} = -\mathbb{E}_{x' \in \mathcal{X}'}\mathcal{D}(x') + \mathbb{E}_{\hat{x} \in \hat{\mathcal{X}}}\mathcal{D}(\mathcal{G}(\hat{x})) + w_{\mathcal{D}}\mathbb{E}_{\hat{x} \in \hat{\mathcal{X}}, x' \in \mathcal{X}'}\nabla_{\alpha x' + (1-\alpha)\hat{x}}\mathcal{D}(\alpha x' + (1-\alpha)\hat{x}),$$

$$L_{\mathcal{G}_x} = \mathbb{E}_{\hat{x} \in \hat{\mathcal{X}}}[L_1(\mathcal{G}(\hat{x}), \hat{x}) + \text{MSE}(\mathcal{G}(\hat{x}), \hat{x}) + \text{LPIPS}(\mathcal{G}(\hat{x}), \hat{x})],$$

$$L_{\mathcal{G}_D} = -w_{\mathcal{G}}\mathbb{E}_{\hat{x} \in \hat{\mathcal{X}}}\mathcal{D}(\mathcal{G}(\hat{x})), \quad L_{\mathcal{G}} = L_{\mathcal{G}_D} + w_x L_{\mathcal{G}_x}.$$

The notations are the same as these in the watermark removal attack. It is easy to find that for both types of attacks, the training framework can be seen as a *unified* one, because the adversary only needs to replace $x'$ with $\hat{x}$ or replace $\hat{x}$ with $x'$, to switch to another attack.

## 5 EVALUATIONS

### 5.1 EXPERIMENT SETUP

**Datasets.** We mainly consider two datasets: CIFAR-10 and CelebA (Liu et al., 2015). CIFAR-10 contains 50,000 training images and 10,000 test images with a resolution of 32*32. CelebA is a celebrity faces dataset, which contains 162,770 images for training and 19,867 for testing, resized at a resolution of 64*64 in our experiments. We randomly split the CIFAR-10 training set into two disjoint parts, one of which is to train the service provider's model and another is used by the adversary. Similarly, we randomly pick 100,000 images for the service provider and 10,000 images for the adversary from the CelebA training set. Furthermore, we also consider a more complex dataset with high resolution (256*256), LSUN (Yu et al., 2015). Furthermore, we also collect some generated images from Stable Diffusion (Rombach et al., 2022) to verify the effectiveness of our method in more complex situations. Details can be found in Appendix F. To enhance the connections with AIGC, we evaluate our method on generative model generated images in Section 5.4 and Appendix H, in which we consider images generated by GANs, conditional diffusion models, and popular Stable Diffusion models.

**Watermarking Schemes.** Considering the watermark's expandability to multiple users, we mainly adopt the post hoc manner, i.e., adding user-specific watermarks to the generated images. We adopt StegaStamp (Tancik et al., 2020), a state-of-the-art and robust method for embedding bit strings into given images, which is proved to be the most effective watermarking embedding method against various removal attacks (Zhao et al., 2023a). **On the other hand, watermarking schemes, such as RivaGAN (Zhang et al., 2019) and SSL (Fernandez et al., 2022), have been shown to be not robust (Zhao et al., 2023a). Therefore, we only consider breaking watermarking schemes,**

---

[3]We slightly modify the discriminator loss for large-resolution images to stabilize the training process. Details are in Appendix A.

| # of Samples (bit length = 32bit) | Original | | | | | Watermark Remove | | | | | Watermark Forge | | | | |
|---|---|---|---|---|---|---|---|---|---|---|---|---|---|---|---|
| | Bit Acc | FID | PSNR | SSIM | CLIP | Bit Acc↓ | FID↓ | PSNR↑ | SSIM↑ | CLIP↑ | Bit Acc↑ | FID↓ | PSNR↑ | SSIM↑ | CLIP↑ |
| 10 | | | | | | 49.98% | 46.90 | 23.19 | 0.81 | 0.83 | 72.64% | 12.27 | 22.43 | 0.89 | 0.91 |
| 50 | 100.00% | 4.14 | 30.69 | 0.94 | 0.96 | 53.31% | 19.74 | 24.47 | 0.87 | 0.86 | 83.18% | 11.89 | 28.37 | 0.94 | 0.93 |
| 100 | | | | | | 53.27% | 14.30 | 25.51 | 0.89 | 0.87 | 93.47% | 12.43 | 26.57 | 0.92 | 0.91 |

Table 5: Few-shot generalization ability of `Warfare` on unseen watermarks on CelebA.

| Methods | WGAN-div | | | | | | EDM | | | | | |
|---|---|---|---|---|---|---|---|---|---|---|---|---|
| | Original | | Watermark Remove | | Watermark Forge | | Original | | Watermark Remove | | Watermark Forge | |
| | Bit Acc | FID | Bit Acc | FID | Bit Acc | FID | Bit Acc | FID | Bit Acc | FID | Bit Acc | FID |
| $DM_s$ | | | 67.12% | 100.93 | 49.17% | 68.79 | | | 51.03% | 78.08 | 51.14% | 79.75 |
| $DM_l$ | | | **47.16%** | 117.80 | 46.20% | 83.36 | | | 51.69% | 58.39 | 51.31% | 60.00 |
| $VAE_{SD}$ | 99.66% | 60.20 | 67.32% | 45.86 | 49.29% | 19.98 | 99.99% | 8.68 | 49.69% | 28.38 | 49.71% | 26.77 |
| $VAE_C$ | | | 55.11% | 106.94 | 54.07% | 44.59 | | | **48.88%** | 137.81 | 48.94% | 138.19 |
| Warfare | | | 52.12% | 69.88 | **95.72%** | 5.84 | | | 64.56% | 19.58 | **90.75%** | 5.98 |

Table 6: Results of attacking content watermarks from the WGAN-div and EDM.

**which have not been broken before.** We also provide two case studies to explore the prior manner, which directly generates images with watermarks for our case studies. We follow previous works (Fei et al., 2022; Zhao et al., 2023b) to embed a secret watermark to WGAN-div (Wu et al., 2018) and EDM (Karras et al., 2022).

**Baselines.** To the best of our knowledge, `Warfare` is the first work to remove or forge a watermark in images under a pure black-box threat model. Therefore, we consider some potential baseline attack methods under the same assumptions and attacker's capability, i.e., having only watermarked images. These baseline methods can be classified into three groups. (1) Image transformation methods: we consider modifying the properties of the given image, such as resolution, brightness, and contrast. We also consider image compression (e.g., JPEG) and image disruptions (e.g., Gaussian blurring, adding Gaussian noise). (2) Diffusion model methods (Li, 2023): we directly adopt a pre-trained unconditional diffusion model (DiffPure (Nie et al., 2022)) to modify the given image, which does not require to train a diffusion model from scratch and does not need clean images. (3) VAE model methods (Zhao et al., 2023a): we directly adopt two different VAE models. One is from the Stable Diffusion (Rombach et al., 2022), which is named $VAE_{SD}$. Another one is trained on CelebA, which is named $VAE_C$. Specifically, both diffusion models and VAE models are not trained or fine-tuned for watermark removal or forge due to the black-box threat model. We do not adopt guided diffusion models or conditional diffusion models as (Li, 2023) did as well. The results from pre-trained diffusion models are various on different datasets, which will be discussed in Appendix C. Specifically, for watermark removal, the watermarked images are inputs for the attacks; for watermark forge, the clean images are inputs for the attacks.

**Implementation.** We adopt DiffPure (Nie et al., 2022) as the diffusion model used in the second step of `Warfare` **without any fine-tuning**. The diffusion model used in DiffPure depends on the domain of watermarked images. For example, if the watermarked images are human faces from CelebA and FFHQ, we use a diffusion model trained on CelebA. As the adversary does not have any knowledge of the watermarking scheme, it is important to decide which checkpoint should be used in the attack. We provide a simple way to help the adversary select a checkpoint during the training process in Appendix B. More details can be found in Appendix A, including hyperparameters and bit strings.

**Metrics.** To fairly evaluate our proposed `Warfare`, we consider five metrics to measure its performance from different perspectives. To determine the quality of the watermark removal (forging) task, we adopt **Bit Acc**, which can be calculated as $\text{Bit Acc}(m, m') = \frac{|m| - \text{HD}(m, m')}{|m|} \times 100\%$, where $\text{HD}(\cdot, \cdot)$ is the Hamming Distance. If $\text{Bit Acc}(m, m') \geq \tau$, verification will pass. Otherwise, it will fail. In our experiments, $\tau = 80\%$. To evaluate the quality of the images generated by `Warfare` and the baselines, we adopt the Fréchet Inception Distance (FID) (Heusel et al., 2017), the peak signal-to-noise ratio (PSNR) (Horé & Ziou, 2010), and the structural similarity index (SSIM) (Horé & Ziou, 2010). Furthermore, we consider the semantic information inside the images, which is evaluated by CLIP (Radford et al., 2021). For the FID, PSNR, SSIM, and CLIP scores, we compute the results between clean images and watermarked images for the watermarking scheme, and between clean images and images after removal or forge attacks. For watermark removal, a lower bit accuracy is better. For watermark forging, a higher bit accuracy is better. For all tasks, a higher PSNR, SSIM, and CLIP score is better. And a lower FID is better.

## 5.2 ABLATION STUDY

In this section, we explore the generalizability of our proposed `Warfare` under the views of the length of the embedding bits and the number of collected images. In Table 2, we show the results of `Warfare` at different lengths of embedded bits. The results indicate that `Warfare` is robust for different secret message lengths. Specifically, when the length of the embedded bits increases,

`Warfare` can still achieve good performance on watermark removing or forging and make the transferred images keep high quality and maintain semantic information. In Table 1, we present the results when the adversary uses the different numbers of collected images as his training data. The results indicate that even with limited data, the adversary can remove or forge a specific watermark without harming the image quality, which proves that our method can be a real-world threat. Therefore, our proposed `Warfare` has outstanding flexibility and generalizability under a practical threat model. We further prove its extraordinary few-shot generalizability for unseen watermarks in Section 5.3.

## 5.3 RESULTS ON POST HOC MANNERS

Here, we focus on post hoc manners, i.e., adding watermarks to AIGC with an embedding model. Because the post hoc watermarking scheme can freely change the embedding watermarks, we evaluate `Warfare` under few-shot learning to show the capability of adapting to unseen watermarks.

**Results on CelebA.** We consider two different lengths of the embedding bits, i.e., 32-bit and 48-bit. Furthermore, we do not consider the specific coding scheme, including the source coding and the channel coding. Tables 3 and 4 compare `Warfare` and the baseline methods on the watermark removal task and the watermark forging task, respectively. We notice that the watermark embedding method is robust against various image transformations. Using image transformations cannot simply remove or forge a specific watermark in the given images[4]. For methods using diffusion models, we consider two settings, i.e., adding large noise to the input ($DM_l$) and adding small noise to the input ($DM_s$). Especially, we use the same setting as $DM_l$ in the second step of `Warfare` to generate images. Although diffusion models can easily remove the watermark from the given images under both settings, the generated images are visually different from the input images, causing a low PSNR, SSIM, and CLIP score. Furthermore, the FID indicates that the diffusion model will cause a distribution shift compared to the clean dataset. Nevertheless, we find that $DM_l$ and $DM_s$ can maintain high image quality while successfully removing watermarks on other datasets, which we discuss in Appendix C. The results make us reflect on the generalizability of diffusion models on different datasets and watermarking schemes. However, evaluating all accessible diffusion models on various datasets and watermarking schemes will take months. Therefore, we leave it as future work to deeply study the diffusion models in the watermarking removal task. On the other hand, forging a specific unknown watermark is non-trivial and impossible for both image transformation methods and diffusion models.

Our `Warfare` gives an outstanding performance in both tasks and maintains good image quality as well. However, we notice that as the length of the embedded bit string increases, it becomes more challenging to forge or remove the watermark. That is the reason that under 48-bit length, our `Warfare` has a little performance drop on both tasks with respect to bit accuracy and image quality. We provide visualization results in the following content to prove images generated by `Warfare` are still visually close to the given image under a longer embedding length. More importantly, `Warfare` is time-efficient compared to diffusion model methods. The results are in Appendix D.

**Few-Shot Generalization.** In real-world applications, large companies can assign a unique watermark for every account or change watermarks periodically. Therefore, it is important to study the few-shot power of `Warfare`, i.e., fine-tuning `Warfare` with several new data with an unseen watermark to achieve outstanding watermark removal or forging abilities for the unseen watermark. In our experiments, we mainly consider embedding a 32-bit string into clean images. Then, we fine-tune the model in Table 3 to fit new unseen watermarks. In Table 5, we present the results under 10, 50, and 100 training data for watermark removal and forging. The results indicate that the watermark removal task is much easier than the watermark forging task. Furthermore, with more accessible data, both bit accuracy and image quality can be improved. It is worth noticing that, even with limited data, `Warfare` can successfully remove or forge an unseen watermark and maintain high image quality. The results prove that our proposed method has strong few-shot generalization power to meet practical usage.

**Visualization.** To better compare the image quality of `Warfare` with other baselines, we show the visualization results in Appendix G. Specifically, both $DM_s$ and $DM_l$ will change the semantic information in inputs. `Warfare` can keep the image details in the watermark removal and forging tasks. Furthermore, when comparing the differences between clean and watermarked images, we find that `Warfare` can produce a similar residual as the watermark embedding model, which means that

---

[4]We omit the results with image transformations in the following tables to save space.

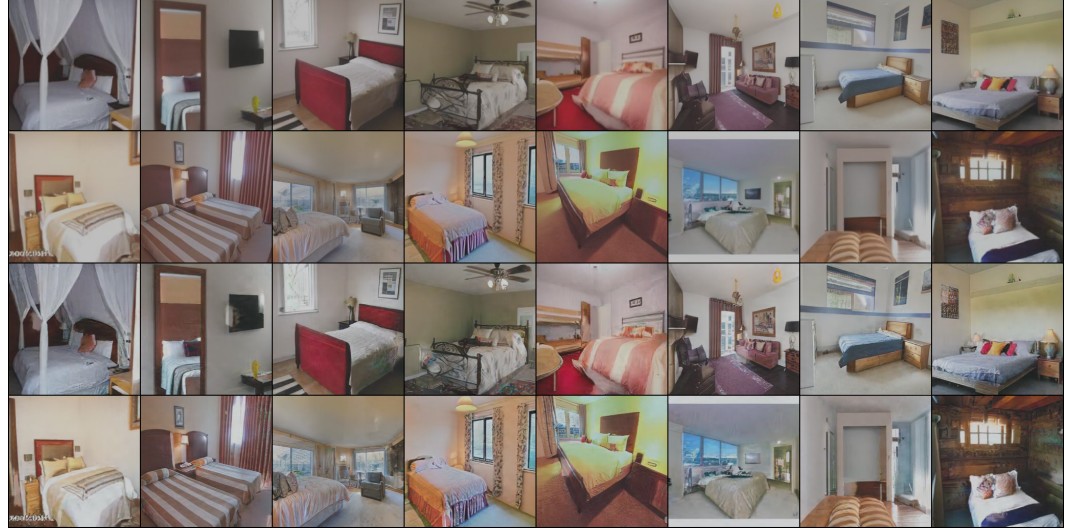

Figure 2: Clean images and outputs from `Warfare`. The top two rows are clean images.

`Warfare` can learn the embedding information during the training process. **Note that the generated images by `Warfare` will be improved with bigger model structures and training data.** Because our principal aim is to prove the effectiveness of our method, the generator we use is quite simple. The structure of our generator is several cascaded Residual blocks, which can be replaced with more advanced structures, such as StyleGAN (Karras et al., 2019). Therefore, **we believe that it will be easy to further improve the image quality.**

## 5.4 RESULTS ON PRIOR MANNERS

We focus on prior methods, i.e., directly embedding watermarks into generative models. We follow the previous methods (Fei et al., 2022) and (Zhao et al., 2023b) to embed a secret bit string into a WGAN-div and an EDM as a watermark, respectively. Therefore, all generated images contain a pre-defined watermark, but we cannot have the corresponding clean images. That is to say, we cannot obtain the PSNR, SSIM, and CLIP scores as previously. So, we only evaluate the FID and the bit accuracy in our experiments. Specifically, we train the WGAN-div with 100,000 watermarked images randomly selected from the training set of CelebA. We directly use the models provided by (Zhao et al., 2023b), which are trained on FFHQ embedded with a 64-bit string. For `Warfare`, we use the WGAN-div and EDM to generate 10,000 samples as the accessible data. In Table 6, we show the results of different attacks to remove or forge the watermark. First, we find that embedding a watermark in the generative model will cause the generated images to have a different distribution from the clean images, making the FID extremely high. Second, EDM can generate high-quality images even under watermarking, causing a lower FID. However, we find that the embedded watermark by (Zhao et al., 2023b) is less robust, which can be removed by blurring and JPEG compression. It could be because they made some trade-off between image quality and robustness. For both, `Warfare` can successfully remove and forge the specific watermark in the generated images and maintain the same image quality as the generative model. The visualization results can be found in Appendix G.

Overall, `Warfare` can pass the watermark verification process for the watermarking forging attack. `Warfare` will make the image fail to pass the watermark verification process for the watermark removal attack. `Warfare` is a practical threat for both post hoc methods and prior methods.

## 5.5 LARGE-RESOLUTION AND COMPLEX IMAGES

To better show the practical usage on large and complex images, in Figure 2, we compare the images from LSUN before and after `Warfare`, in which our target is to forge a specific watermark. It is difficult for human eyes to determine which clean images are, which shows that `Warfare` can maintain impressive image quality even for complex and large-resolution images. Clearly, `Warfare` is still effective for large-resolution and complex images. We obtain about 80% bit accuracy and 44 FID for the forging attack. More analysis and numerical results can be found in Appendix F.

## 5.6 Generalize to AIGC and Out-of-Distribution Images

We first extend `Warfare` to latent diffusion models. We use only 100 images generated by Stable Diffusion 1.5 watermarked by the post hoc manner to fine-tune the `Warfare` models in Table 4. The reason that we adopt the post hoc watermarking manner is that it can easily assign different watermarks for users, which cannot be achieved by the prior methods. Then, we evaluate the watermark attacks on 1,000 generated images by Stable Diffusion 1.5. For watermark removal, the bit accuracy decreases from 99.98% to 51.86% with FID 23.53. For watermark forging, the bit accuracy is 80.07% with FID 39.38. Although our results are based on few-shot learning, instead of directly training on massive images generated by Stable Diffusion, the results still show the generalizability of `Warfare`. Second, we evaluate the zero-shot capability of `Warfare` with Tiny ImageNet for models from Table 4. The bit accuracy for watermark removal is about 90% and about 70% for watermark forging. Although the zero-shot capability is limited, it is easy to improve the performance with 100 samples to fine-tune the model, obtaining about 50% bit accuracy for removal and 90% bit accuracy for forging. Therefore, `Warfare` can easily be generalized to other domains.

## 6 Potential Defenses for Service Providers

Although `Warfare` is an effective method for removing or forging a specific watermark in images, there are some possible defense methods against our attack. First, large companies can assign a group of watermarks to an account to identify the identity. When adding watermarks to images, the watermark can be randomly selected from the group of watermarks, which can hinder the adversary from obtaining images containing the same watermark. However, such a method requires a longer length of embedded watermarks to meet the population of users, which will decrease image quality because embedding a longer watermark will damage the image. We provide a case study to verify such a defense. In our implementation, we choose to use two bit strings for one user, i.e., $m_1$ is '100010001000100010001000' and $m_2$ is '11100011101010101000010000001011'. Note that the Hamming Distance between $m_1$ and $m_2$ is 12, which means that there are 12 bits in $m_1$ and $m_2$ are different. We assume that $m_1$ and $m_2$ will be used with equal probability. Therefore, half of the collected data contain $m_1$ and others contain $m_2$. We evaluate `Warfare` on this collected dataset. For the watermark removal attack, the bit accuracy for $m_1$ after `Warfare` is 71.04%. And the bit accuracy for $m_2$ after `Warfare` is 64.87%. Note that the ideal bit accuracy after the removal attack is $(32 - 12)/32 * 100\% = 62.50\%$. Therefore, our method can maintain the attack success rate to some degree. For the watermark forging attack, the bit accuracy for $m_1$ after `Warfare` is 87.53%. And the bit accuracy for $m_2$ after `Warfare` is 69.21%. We notice that the ideal bit accuracy for the forging attack is $(32 - 12 + 6)/32 * 100\% = 81.25\%$, which means that 26 bits can be correctly recognized. The results indicate that the generator does not equally learn $m_1$ and $m_2$. We think it is because of the randomness in the training process. On the other hand, the results indicate that such a defense can improve the robustness of the watermark. However, we find `Warfare` can still remove or forge one of the two watermarks. This means that such a defense can only alleviate security problems instead of addressing them thoroughly.

Another defense is to design a more robust watermarking scheme, which can defend against removal attacks from diffusion models. Because `Warfare` requires diffusion models to remove the watermarks. The two methods mentioned above have the potential to defend against `Warfare` but have different shortcomings, such as decreasing image quality, requiring a newly designed coding scheme, and requiring a newly designed robust watermarking scheme. Therefore, `Warfare` will be a threat for future years.

## 7 Limitations and Conclusions

In this paper, we consider a practical threat to AIGC protection and regulation schemes, which are based on the state-of-the-art **robust and invisible** watermarking technologies. We introduce `Warfare`, a unified attack framework to effectively remove or forge watermarks over AIGC while maintaining good image quality. With `Warfare`, the adversary only requires watermarked images without their corresponding clean ones, making it a real-world threat. Through comprehensive experiments, we prove that `Warfare` has strong few-shot generalization abilities to fit unseen watermarks, which makes it more powerful. Furthermore, we show that `Warfare` can easily replace a watermark in the collected data with another new one, in Appendix E.

We discuss the potential usage of `Warfare` for larger-resolution and more complex images, in real-world scenarios. Further improvement over `Warfare` is probable with more advanced GAN structures and training strategies.

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

| Experiment | Watermark Remove | | Watermark Forge | |
|---|---|---|---|---|
| | $w_{\mathcal{G}}$ | $w_x$ | $w_{\mathcal{G}}$ | $w_x$ |
| CIFAR-10 4bit | 500 | 10 | 500 | 5 |
| CIFAR-10 8bit | 800 | 15 | 500 | 10 |
| CIFAR-10 16bit | 500 | 40 | 150 | 40 |
| CIFAR-10 32bit | 100 | 40 | 100 | 40 |
| CIFAR-10 5000 data | 800 | 15 | 500 | 10 |
| CIFAR-10 10000 data | 800 | 15 | 600 | 20 |
| CIFAR-10 15000 data | 500 | 15 | 500 | 10 |
| CIFAR-10 20000 data | 800 | 15 | 500 | 15 |
| CIFAR-10 25000 data | 800 | 15 | 500 | 10 |
| CelebA 32bit | 10 | 120 | 1 | 10 |
| CelebA 48bit | 10 | 200 | 1 | 10 |
| Few-Shot 10 Images | 10 | 200 | 1 | 10 |
| Few-Shot 50 Images | 10 | 200 | 1 | 10 |
| Few-Shot 100 Images | 10 | 200 | 1 | 10 |
| WGAN-div | 10 | 120 | 1 | 10 |
| EDM | 1 | 10 | 100 | 1 |

Table 7: Hyperparameter settings in our experiments for watermark removal and watermark forging.

## A  EXPERIMENT SETTINGS

**Model Structures.** For CIFAR-10 and CelebA, we choose different architectures for generators and discriminators to stabilize the training process. Specifically, when training models on CIFAR-10, we use the ResNet-based generator architecture (Zhu et al., 2017) with 6 blocks. As the CelebA images have higher resolution, we use the ResNet-based generator architecture (Zhu et al., 2017) with 9 blocks. For the discriminators, we use a simple model containing 4 convolutional layers for CIFAR-10. And for CelebA, a simple discriminator cannot promise a stable training process. Therefore, we use a ResNet-18 (He et al., 2016). To improve the quality of generated images, we follow the residual training manner, that is, the output from the generators will be added to the original input.

**Hyperparameters.** We use different hyperparameters for CIFAR-10 and CelebA, respectively. When training models on CIFAR-10, we use RMSprop as the optimizer for both the generator and the discriminator. The learning rate is 0.0001, and the batch size is 32. We set $w_{\mathcal{D}} = 10$, and the total number of training epochs is 1,000. We update the generator's parameters after 5 times of updating of the discriminator's parameters. For CelebA, we adopt Adam as our model optimizer. The learning rate is 0.003, and the batch size is 16. We replace the discriminator loss with the one from StyleGAN (Karras et al., 2019) with $w_{\mathcal{D}} = 5$, and the total number of training epochs is 1,000. We update the generator's parameters after updating the discriminator's parameters. We present $w_{\mathcal{G}}$ and $w_x$ in Table 7 used in our experiments. We choose the best model based on the image quality.

**Baseline Settings.** For image transformation methods, we mainly adopt `torchvision` to implement attacks. To adjust brightness, contrast, and gamma, the changing range is randomly selected from 0.5 to 1.5. To adjust the hue, the range is randomly selected from -0.1 to 0.1. For center-cropping, we randomly select the resolution from 32 to 64. For the Gaussian blurring, we randomly choose the Gaussian kernel size from 3, 5, and 7. For adding Gaussian noise, we randomly choose $\sigma$ from 0.0 to 0.1. For JPEG compression, we randomly selected the compression ratio from 50 to 100. When evaluating the results of image transformation methods, we run multiple times and use the average results. For diffusion methods $DM_l$, we set the sample step as 30 and the noise scale as 150. For diffusion methods $DM_s$, we set the sample step as 200 and the noise scale as 10. Specifically, we use $DM_l$ in the second step of `Warfare`. Considering using diffusion models to generate images is very time-consuming, we randomly select 1,000 images from the test set to obtain the results for diffusion models.

**Embedded Bits.** In Table 8, we list the bit strings embedded in the images in our experiments.

## B  SELECT A CORRECT CHECKPOINT

It is important to choose the correct checkpoint because it is closely associated with the attack performance. However, when the adversary does not have any information about the watermarking

| Experiment | Bit String |
|---|---|
| CIFAR-10 4bit | 1000 |
| CIFAR-10 8bit | 10001000 |
| CIFAR-10 16bit | 1000100010001000 |
| CIFAR-10 32bit | 10001000100010001000100010001000 |
| CelebA 32bit | 10001000100010001000100010001000 |
| CelebA 48bit | 100010001000100010001000100010001000100010001000 |
| Few-Shot | 111000111010101010000100000001011 |
| WGAN-div | 10001000100010001000100010001000 |
| EDM | 0100010001000010111010101111111100111010000011111011010101100000000 |

Table 8: Selected bit strings in our experiments.

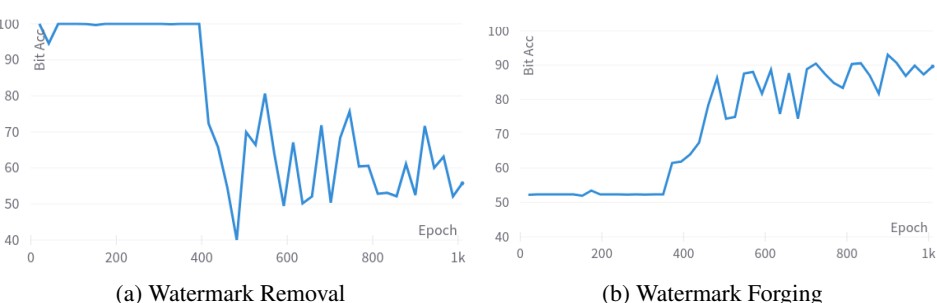

(a) Watermark Removal          (b) Watermark Forging

Figure 3: Bit Acc for different tasks during the training stage on CelebA.

scheme, it is unavailable to determine the best checkpoint with Bit Acc as metrics. However, after plotting the bit accuracy in Figure 3, we find that the performances of different checkpoints in the later period are close and acceptable for a successful attack under the Bit ACC metrics. Therefore, we choose the best checkpoint from the later training period based on the image quality metrics, including the FID, SSIM, and PSNR, in our experiments. It is to say, our selection strategy does not violate the threat model, where the adversary can only obtain watermarked images.

Specifically, as training GANs are challenging, we applied several approaches to stabilize the training process, improve the performance, and ease the usage. First, we adopt a residual manner in GAN structure, as introduced in Appendix A. We increase or decrease the model size based on the resolution of input images. It helps us to obtain outputs with higher quality. Second, we use the discriminator loss from StyleGAN to further stabilize the training process of the discriminator. We further adopt alternate optimization strategies to avoid overfitting of the discriminator. Third, our training script supports Exponential Moving Average (EMA), which is a widely used trick in GAN training. With the above methods, we train our GANs more smoothly and stably. As shown in Figures 3 and 5, the performance of GANs is relatively stable. Besides, we will provide experiment code to make these results reproducible.

## C  DIFFUSION MODELS FOR WATERMARK REMOVAL

In our experiments, we find that the pre-trained diffusion models will not promise a similar output as the input image without the guidance on CelebA. However, when we evaluate the diffusion models on another dataset, LSUN-bedroom (Yu et al., 2015), we find that even under a very large noise scale, the output of the diffusion model is very close to the input image, and the watermark has been successfully removed. The visualization results can be found in Figure 4, where we use 30 sample steps and 150 noise scales for $DM_l$ and use 200 sample steps and 10 noise scales for $DM_s$, which are the same as the settings on CelebA. The numerical results in Table 9 prove that the diffusion model can maintain high image quality under large inserted noise.

We think the performance differences on CelebA and LSUN are related to the resolution and image distribution. Specifically, images in CelebA are 64 * 64 and only contain human faces. The diversity of faces is not too high. However, images in LSUN are 256 * 256 and have different decoration styles, illumination, and perspective, which means the diversity of bedrooms is very high. Therefore, transforming an image into another image in LSUN is more challenging than doing that in CelebA. This could be the reason that diffusion models cannot produce an output similar to that of CelebA. This limitation is critical for an attack based on diffusion models. Therefore, we appeal to comprehensively evaluate the performance of the watermark removal task for various datasets.

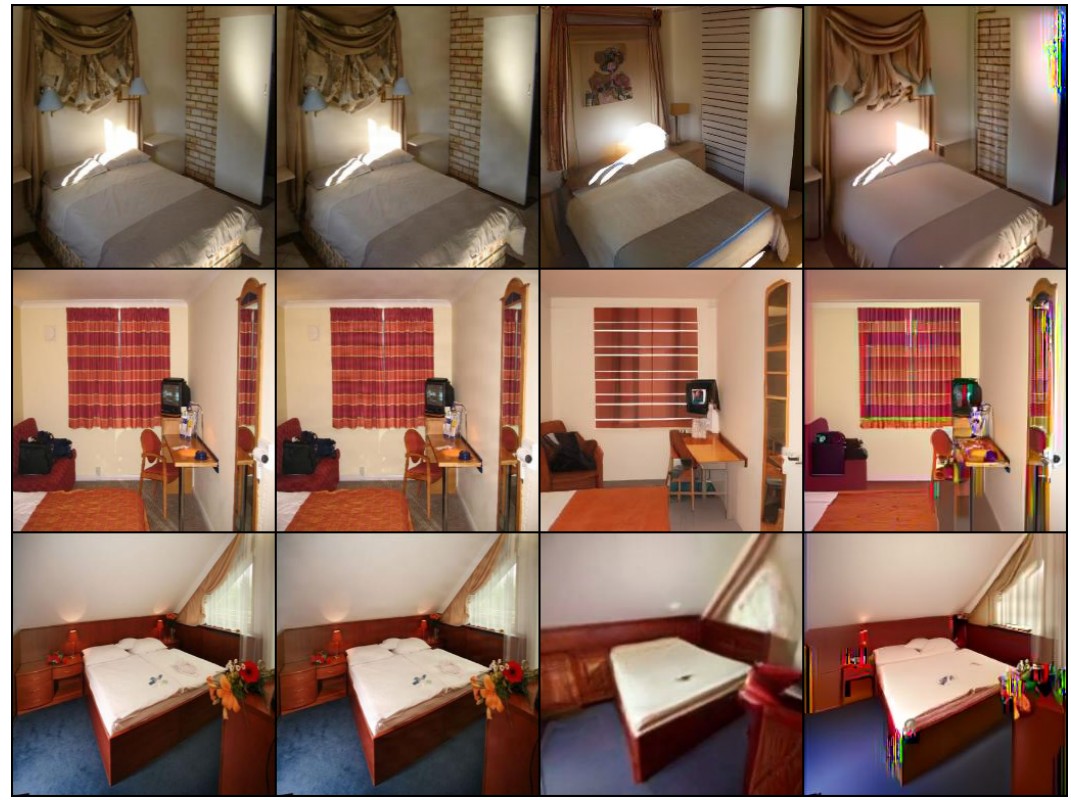

Figure 4: The first column is clean images. The second is watermarked images. The third is the output of $\text{DM}_l$. The fourth is the output of $\text{DM}_s$.

| Diffusion Model Setting (bit length = 32bit) | | Original | | | | | Watermark Remove | | | | |
|---|---|---|---|---|---|---|---|---|---|---|---|
| Sample Step | Noise Scale | Bit Acc | FID | PSNR | SSIM | CLIP | Bit Acc | FID | PSNR | SSIM | CLIP |
| 30 | 150 | | | | | | 51.81% | 75.52 | 20.15 | 0.58 | 0.88 |
| 50 | 150 | | | | | | 51.50% | 84.14 | 18.92 | 0.55 | 0.86 |
| 100 | 150 | | | | | | 50.47% | 95.27 | 16.69 | 0.49 | 0.83 |
| 200 | 10 | 100.00% | 10.67 | 39.49 | 0.98 | 0.99 | 56.16% | 73.01 | 22.11 | 0.72 | 0.84 |
| 200 | 30 | | | | | | 53.03% | 98.00 | 19.37 | 0.59 | 0.80 |
| 200 | 50 | | | | | | 53.81% | 108.71 | 17.63 | 0.52 | 0.78 |

Table 9: Numerical results of watermark removal with diffusion models under different noise scales and sample steps.

The above results prove that a pre-trained diffusion model alone can remove watermarks. However, the limitation that the generation quality is unstable, unreliable, and changing with different data distribution is also clear. Besides, watermark forging is impossible with a pre-trained diffusion model. Therefore, we propose to use GANs in our method, building a unified framework for watermark removal and forging. Our method is proved to achieve a better result with lower FID and effectively build a unified framework for both types of attacks.

## D    TIME COST VS DIFFUSION MODELS

To compare the time cost for generating one image with a given one, we record the total time cost for 1,000 images on one A100. The batch size is fixed to 128. For $\text{DM}_l$, the total time cost is 5,231.72 seconds. For $\text{DM}_s$, the total time cost is 2325.01 seconds. For Warfare, the total time cost is **0.46** seconds. Therefore, our method is very fast and efficient.

We evaluate the time cost of attacking the Stable Signature watermarking scheme on 512×512 resolution images with 8 A6000 GPUs. During the data preprocessing phase, we employ a pre-trained diffusion model to remove watermarks for 10,000 images generated by the watermarked Stable

| Method | Data Preprocessing | GAN Training | Inference | Total |
|---|---|---|---|---|
| Diffusion-base | - | - | 42.4 hrs | 42.4 hrs |
| Warfare | 42.4 hrs | Remove: 4.7 hrs
Forge: 1.3 hrs | 1.84 s | Remove: 47.1 hrs
Forge: 43.7 hrs |

Table 10: Time cost of attacking the Stable Signature watermarking scheme on 512×512 resolution images. We evaluate the time cost when attacking 10,000 images. hrs stands for hours. s stands for seconds.

Diffusion 2.1, taking a total runtime of 42.4 hours. In the GAN training phase, we achieve a forging accuracy of 99% in just 5 epochs, with a total runtime of 1.3 hours, and achieve a removal accuracy of 49% in 18 epochs, with a total runtime of 4.7 hours. During inference, it takes 1.84 seconds for our GAN to forge or remove the watermark of 10,000 images. Therefore, compared with diffusion-based method, our method brings performance improvement with about 1.3 (4.7) hours additional time overhead to forge (remove) watermark. The results can be found in Table 10. Notably, with the few-shot generalization abilities of our method, an attacker can fine-tune a pre-trained GAN using only 10 to 100 samples to remove or forge different watermarks, reducing data preprocessing and GAN training costs by 99%. Therefore, our method has better scalability, generalizability and efficiency in a long-term evaluation.

## E    REPLACE A WATERMARK WITH NEW ONE

We further consider another attack scenario, where the adversary wants to replace the watermark in the collected images with one specific watermark used by other users or companies. In this case, the adversary first trains a generator $G_r$ to remove the watermark in the collected image $x$. Then, the adversary trains another generator $G_f$ to forge the specific watermark. Finally, to replace the watermark in $x$ with the new watermark, the adversary only needs to obtain $x' = G_f(G_r(x))$. We evaluate the performance of Warfare in this scenario on CelebA. Specifically, $G_r$ is the generator in our few-shot experiment. And $G_f$ is the generator in our CelebA 32bit experiment. It is to say that the existing watermark in the collected images is "11100011101010101000010000001011", and the adversary wants to replace it with "1000100010001 0001000100010001000". The details can be found in our main paper. As for the results, we calculate PSNR, SSIM, CLIP score, and FID between $x'$ and clean images. And we also compute the bit accuracy of $x'$ for the new watermark. The FID is 18.67. The PSNR is 24.97. The SSIM is 0.90. The CLIP score is 0.92. And the bit accuracy is **98.86%**. The results prove that Warfare can easily replace an existing watermark in the images with a new watermark.

## F    LARGE-RESOLUTION AND COMPLEX IMAGES

We focus on CelebA in our main paper, which contains human faces in a resolution of 64 * 64. In this part, we illustrate the results of our method on larger resolution and more complex images. To evaluate our method on such images, LSUN-bedroom (Yu et al., 2015) is a good choice, in which the image resolution is 256 * 256. Similarly to the CelebA experiment settings, we randomly select 10,000 images for Warfare, and the bit length is 32. As watermark removal is easy to do with only diffusion models, forging is more challenging and critical. Therefore, we aim to forge a specific watermark on the clean inputs.

In Figure 5, we illustrate the bit accuracy during the training stage of Warfare. Although accuracy increases with increasing training steps, we find that it is difficult to achieve accuracy over 80%. If we increase the number of training steps, the accuracy will be stable around 75%. While Warfare is still effective for large-resolution and complex images, we think its ability is constrained, due to the limited training data and a small generator structure. Our future work will be to improve its effectiveness for more complex data.

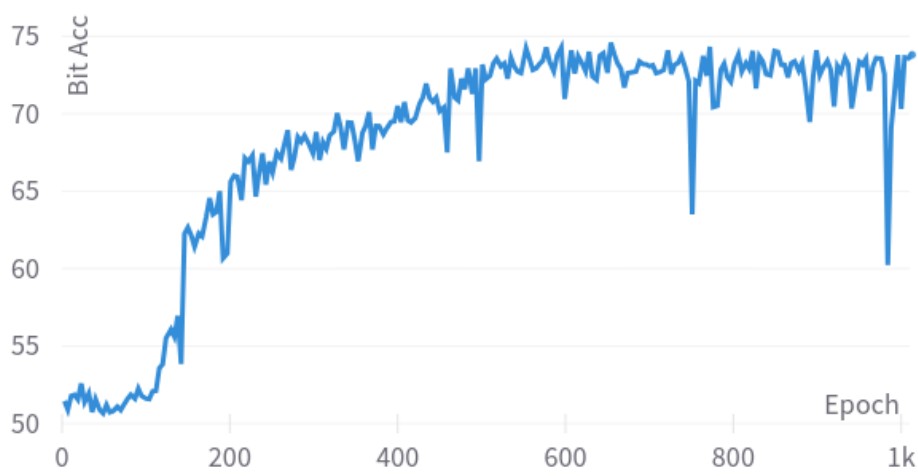

Figure 5: Bit Acc with training epoch increasing.

## G    OTHER VISUALIZATION RESULTS

In this section, we show the other visualization results in our experiments. First, we show Figure 6 in a larger resolution. In Figure 7, we present the visualization results for the few-shot experiments. The results indicate that with more training samples, image quality can be improved. And, even with a few samples, `Warfare` can learn the embedding pattern. In Figure 8, we show the visualization results of WGAN-div and EDM, respectively. The attack goal is to forge a specific watermark. In Figure 9, we present the high-resolution images for LSUN to prove the effectiveness of `Warfare` on larger and more complex photos. In Figure 10, we present the high-resolution images generated by Stable Diffusion 1.5 to show the generalizability of `Warfare` for AI-generated content based on advanced generative models. The results indicate that `Warfare` can generate images with the specific watermark, keeping high quality simultaneously.

## H    WARFARE ON AIGC DATASET

| Method | Origin | | Watermark Remove | | Watermark Forge | |
|---|---|---|---|---|---|---|
| | Bit Acc | FID | Bit Acc | FID | Bit Acc | FID |
| DM | | | **48.29%** | 8.77 | 46.94% | 5.71 |
| VAE | 100% | 7.65 | 52.69% | 8.72 | 48.78% | 2.94 |
| Warfare | | | 49.22% | **8.07** | **99.08%** | **0.78** |

Table 11: Results of attacking Stable Signature on Stable Diffusion 2.1.

Besides the experiments in our paper, we add new results, attacking Stable Signature (Fernandez et al., 2023) in our revision. Stable Signature is a prior watermarking method. The model owner trains a latent decoder for stable diffusion models, which can add a pre-fixed bit string to the generated image. The setups used in the Stable Signature scheme are provided below. We fix the secret key as '111010110101000001010111010011010100010000100111', which is a bit string with length 48. The diffusion model used in Stable Signature is Stable Diffusion 2.1 (SD2.1). During the generation process, we adopt the unconditional generation approach by setting the prompt empty to obtain images with 512*512 resolution. We sample 10,000 watermarked images for our attack method.

For baseline methods, we use a pretrained diffusion model based on ImageNet and the VAE from Stable Diffusion 1.4. Specifically, when using the diffusion model, we set the noise scale as 75 and set the number of sampling step as 15. The setup for the diffusion model is also adopted in our data preprocessing phase. During the processing of GAN training phase, we set the hyperparameters $w_1$=10 and $w_2$=5 for watermark removal attack and set $w_1$=10 and $w_2$=100 for watermark forging attack.

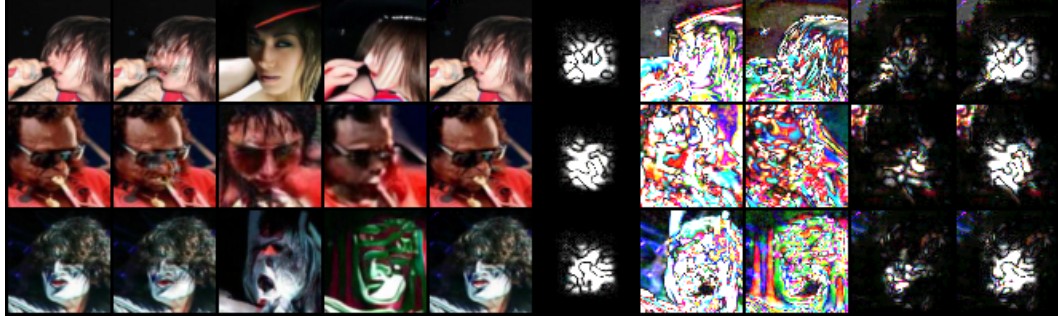

(a) Watermark Removal

(b) Watermark Forging

Figure 6: The first column is clean images. The second is watermarked images. The third is the output of $\mathrm{DM}_l$. The fourth is the output of $\mathrm{DM}_s$. The fifth is the output of `Warfare`. The sixth is the difference between the first and second columns. The seventh is the difference between the first and third columns. The eighth is the difference between the first and fourth columns. The ninth is the difference between the first and fifth columns. The tenth is the difference between the second and fifth columns.

We present the results in Table 11. The results indicate that our method is effective against AIGC watermarking schemes. Even if we train the GAN models on a large resolution dataset, we can obtain models with high performance in both image quality and watermark removal (forging). In Figures 11 and 12, we illustrate the images to visualize the quality. The results prove that `Warfare` can significantly keep the image quality and generate watermarked or unwatermarked images freely.

## I  WARFARE−PLUS WITH HIGHER TIME EFFICIENCY

In `Warfare`, we adopt a pre-trained diffusion model to purify the watermarked data and obtain the mediator images. This brings additional time cost, which reduces the overall time efficiency of our proposed method. To further improve the time efficiency, we propose `Warfare-Plus` by revising the data pre-processing process. We find that the purified images are not essential in our attack framework. Therefore, we directly adopt an open-sourced Stable Diffusion 1.5 (SD1.5) to generate images without conditional prompts as the mediator images. We evaluate the efficiency and effectiveness of `Warfare-Plus` by attacking the aforementioned Stable Signature watermarking scheme under the same configurations. The attack results are presented in Table 12 and the time cost is listed in Table 13. `Warfare-Plus` requires a longer training time to make the GANs converge but needs much less time cost for the data preprocessing process. Compared with `Warfare`, `Warfare-Plus` reduces the total time cost, including data pre-processing, model training, and inference, by 80%∼85%, and keeps good attack performance. In Figures 13 and 14, we show the visualization results of `Warfare-Plus`.

| Method | Origin | | Watermark Remove | | Watermark Forge | |
|---|---|---|---|---|---|---|
| | Bit Acc | FID | Bit Acc | FID | Bit Acc | FID |
| Warfare | 100% | 7.65 | 49.22% | 8.07 | 99.08% | 0.78 |
| Warfare-Plus | | | 49.95% | 8.45 | 97.03% | 1.22 |

Table 12: Results of `Warfare-Plus`, attacking Stable Signature on Stable Diffusion 2.1.

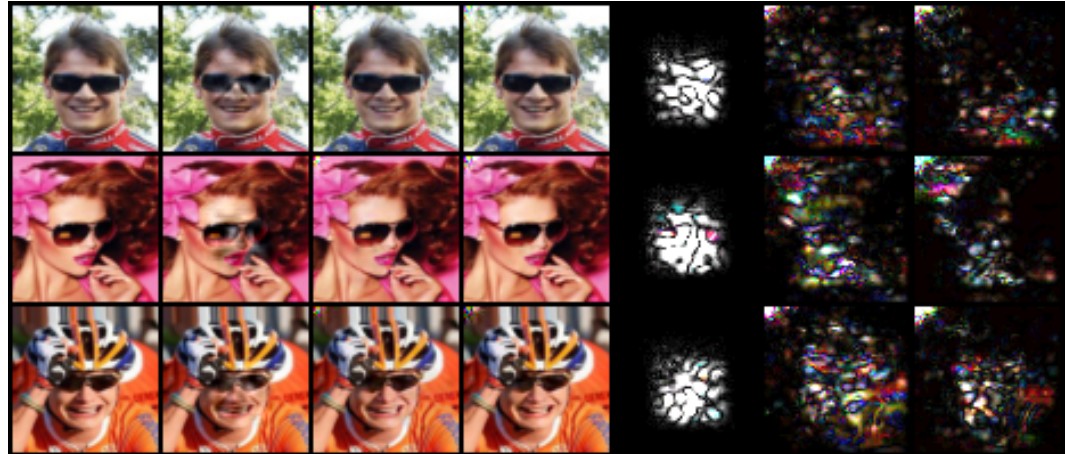

(a) Watermark Removal

(b) Watermark Forging

Figure 7: The first column is clean images. The second is watermarked images. The third is the output of Warfare under the 50-sample setting in the few-shot experiment. The fourth is the output of Warfare under the 100-sample setting in the few-shot experiment. The fifth is the difference between the first and second columns. The sixth is the difference between the first and third columns. The seventh is the difference between the first and fourth columns.

| Method | Data Preprocessing | GAN Training | Inference | Total |
|---|---|---|---|---|
| Warfare | 42.4 hrs | Remove: 4.7 hrs
Forge: 1.3 hrs | 1.84 s | Remove: 47.1 hrs
Forge: 43.7 hrs |
| Warfare-Plus | 1.68 hrs | Remove: 4.96 hrs
Forge: 7.05 hrs | 1.84 s | Remove: 6.64 hrs
Forge: 8.73 hrs |

Table 13: Results of Warfare-Plus by attacking the Stable Signature watermarking scheme on 512×512 resolution images. We evaluate the time cost when attacking 10,000 images. hrs stands for hours. s stands for seconds.

## J    SOCIAL IMPACT

The advent of AI-generated content has ushered in an era marked by unparalleled creativity and efficiency, but this technological leap is not without its ethical and legal ramifications. For example, a very recent case where Taylor Swift's fake photos are circulated on X, which are made by generative models. On the other hand, the Gemini AI model conducted by Google Inc., is believed to generate biased content, by making white famous people black. Clearly, the ethical dilemma lies in recognizing the owner of content, as well as discerning the ethical implications of content manipulation. This resonates not only with the creative industries but extends to broader societal implications, particularly

in the context of misinformation and deepfakes. To mitigate the harm from fake content and biased content and better attribute the owner of the generated content, big companies, like OpenAI and Adobe, have developed and used watermarking methods, such as C2PA, in their products, such as DALL·E 3.

The deployment of content watermarking technologies emerges as a potential solution to safeguard intellectual property in the realm of AI-generated content. However, this introduces its own set of ethical considerations, when considering its robustness. While content watermarking provides a mechanism for tracing the origin of content and protecting the rights of creators, it concurrently raises concerns about potential attacks against such technologies to escape from being watermarked or forge another's watermark.

Significantly, one of the vital parts of the effectiveness of content watermarking technologies is contingent upon their resilience to attacks aimed at their removal or forgery. As shown in our paper, the adversary can manipulate the existing watermarks in the generated content to achieve malicious purposes, including unauthorized use, manipulation of AI-generated content, and framing up others. Addressing these vulnerabilities requires a comprehensive understanding of potential attacks and the development of robust watermarking techniques that can withstand sophisticated adversarial attempts.

Based on our experiments, we can find that the removal or forgery of watermarks not only undermines the protection of intellectual property but also amplifies the risks associated with the misuse of AI-generated content. The malicious alteration of content, coupled with the absence of reliable watermarking, exacerbates the challenges associated with content verification and attribution. Consequently, mitigating the threat of attacks on content watermarks is paramount for ensuring the integrity and trustworthiness of AI-generated content in various domains, including journalism, entertainment, and education. This asks us to develop more advanced content watermarking methods.

Specifically, there are two benefits brought by our attack. First, in Section 6, we prove that the group-watermarking method is promising against `Warfare`. It provides the big companies with a lightweight scheme to improve their current watermarking methods, without developing new models. Second, our attack could become a red-teaming evaluation method to help companies develop more robust and secure watermarking schemes. Developers can adopt our method to test their current watermarking method and conduct specific adjustment to further defend attacks.

In conclusion, we think that the proposed `Warfare` will cause some malicious users to freely make AIGC for commercial use and frame other users by spreading illegal AIGC with forged watermarks. On the other hand, we think besides these negative impacts, our work will encourage others to explore a more robust and reliable watermark for AIGC, which has a positive impact on society. It can be achieved only after we have a deeper study on attacks.

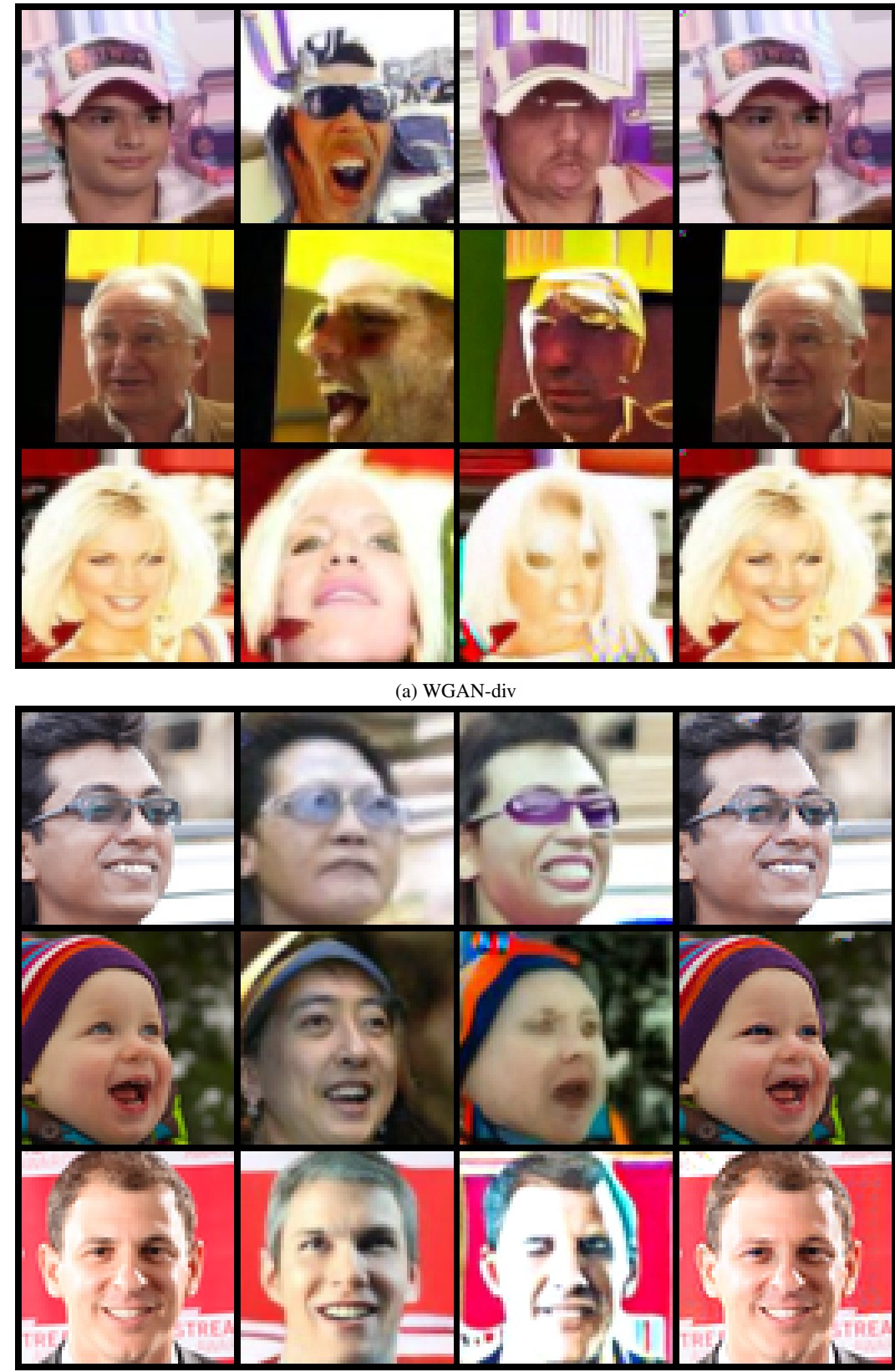

(a) WGAN-div

(b) EDM

Figure 8: Visualization results for prior watermarking methods. The first column is clean images. The second is the output of $\text{DM}_l$. The third is the output of $\text{DM}_s$. The fourth is the output of `Warfare`.

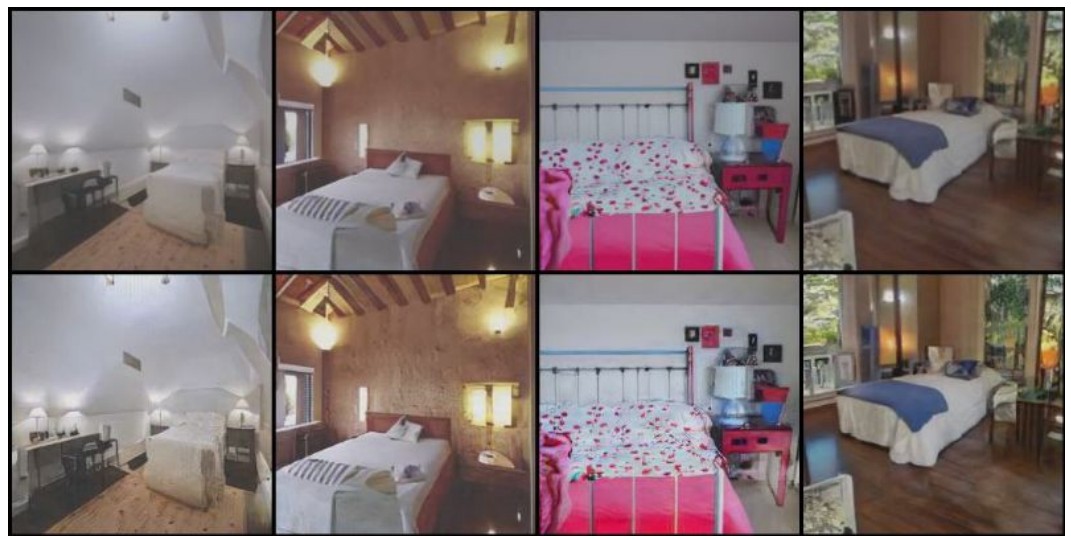

Figure 9: Visualization results of LSUN-bedroom. The first row is clean images. The second is the output of `Warfare`.

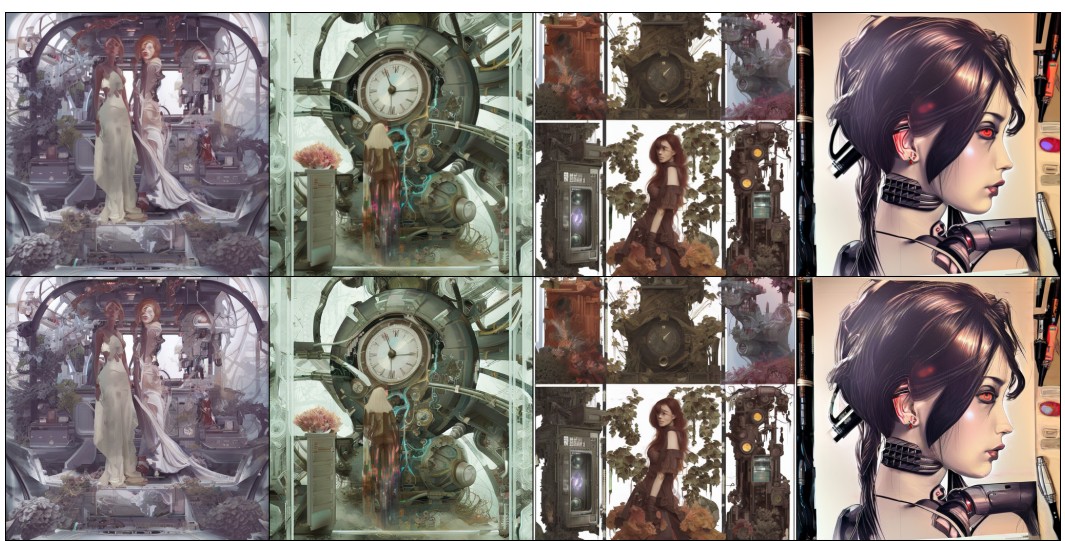

Figure 10: Visualization results of images generated by SD1.5. The first row is clean images. The second is the output of `Warfare`.

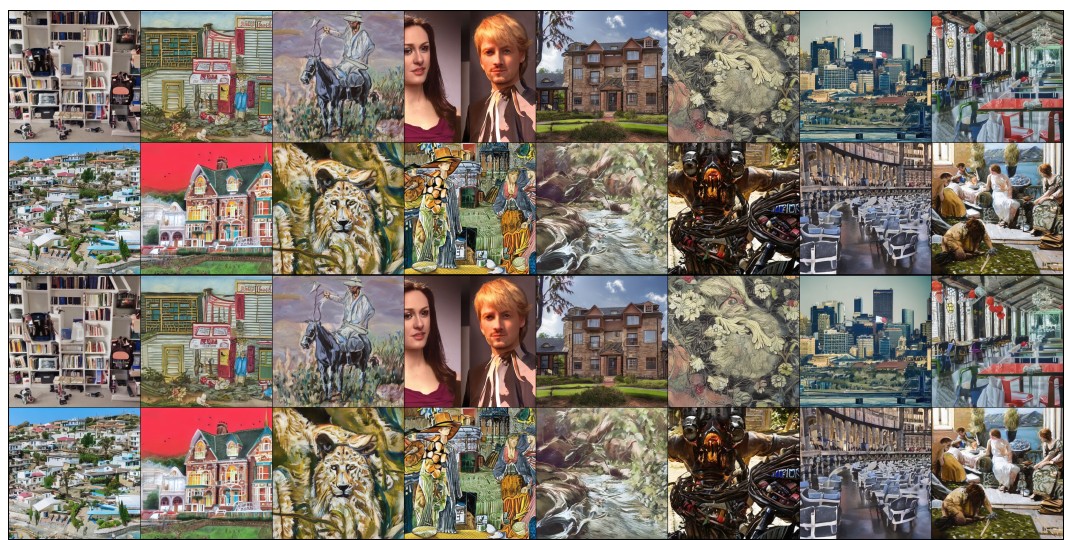

Figure 11: Visualization results of images generated by SD2.1. The first two rows are watermarked images by Stable Signature. The last two rows are the output of `Warfare` to remove the watermark.

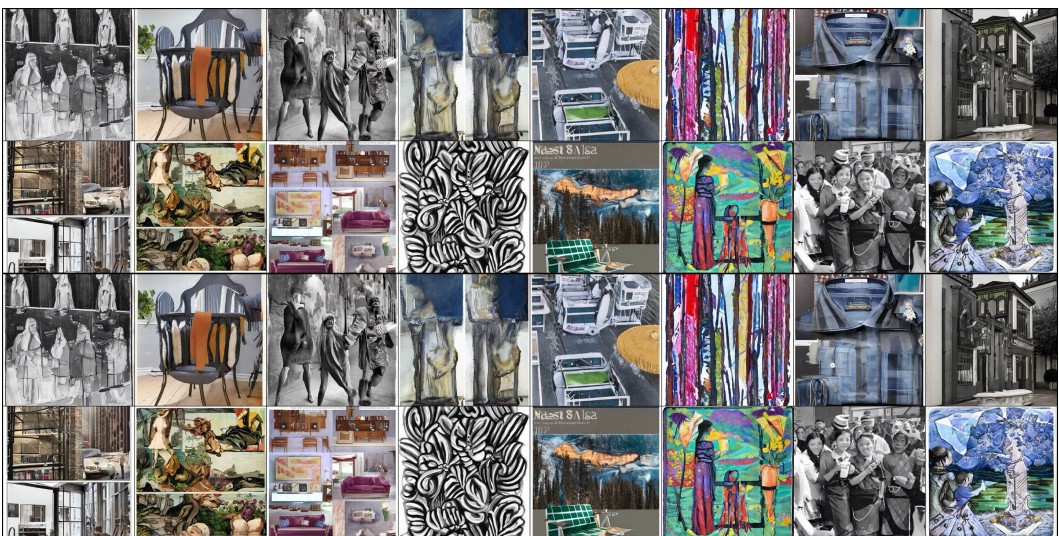

Figure 12: Visualization results of images generated by SD2.1. The first two rows are clean images. The last two rows are the output of `Warfare` to forge the watermark.

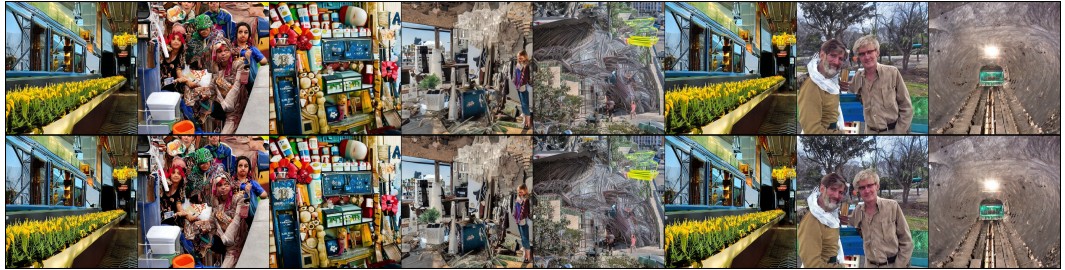

Figure 13: Visualization results of `Warfare-Plus`. The first row is watermarked images by Stable Signature. The last row is the output of `Warfare-Plus` to remove the watermark.

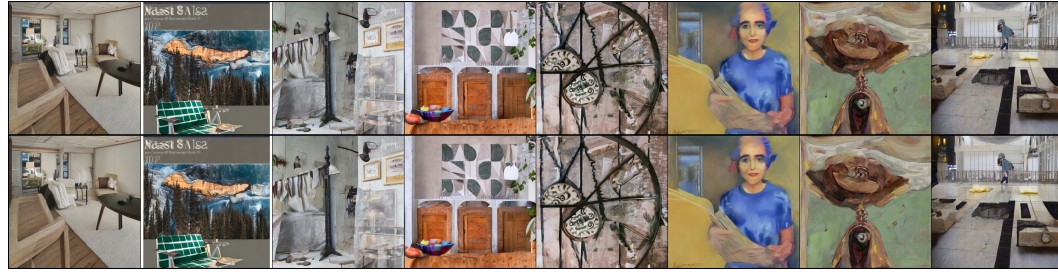

Figure 14: Visualization results of `Warfare-Plus`. The first row is clean images. The last row is the output of `Warfare-Plus` to forge the watermark.

