# OpenReview forum: "Warfare: Breaking the Watermark Protection of AI-Generated Content"
_ICLR.cc/2025/Conference — Submitted to ICLR 2025_

### Official Review · Reviewer_frro · 2024-11-01

**Soundness:** 3
**Presentation:** 3
**Contribution:** 2
**Rating:** 6
**Confidence:** 4

**Summary:**

This paper introduces WARFARE, a strategy to remove or forge invisible image watermarks, by introducing a generative adversarial network based on a user-specific watermarked dataset and its regeneration-attacked counterpart. The method has demonstrated its capability to successfully remove and forge "StegaStamp" watermarks, utilizing the pre-trained diffusion model "DiffPure" without any need for additional fine-tuning. Moreover, the paper highlights that WARFARE performs well even with limited fine-tuning data which indicates strong few-shot generalization ability, achieving significant time efficiency—boasting a speedup of 5,050 to 11,000 times compared to conventional regeneration attacks.

**Strengths:**

This paper is well-written and easy to read. The forging attack threat model is interesting and the solution is novel. The experiment result is clear.

**Weaknesses:**

While I appreciate the novelty in the proposed forging component of this paper, I find the overall contribution to be somewhat incremental. My main concern centers around the assumption regarding the mediator dataset. Specifically, the authors assume that "the mediator dataset can be considered as being drawn from the same 'non-watermarked' distribution, which is distinct from the 'watermarked' distribution." This is a critical assumption underlying why the proposed method is effective. However, whether this assumption holds entirely depends on the interaction between the watermarking technique and the methodology employed to generate the 'non-watermarked' distribution (i.e., the regeneration attack). In my view, this approach inherently relies on the success of a preceding watermark removal attack. The author could benefit from discussing the potential limitations or edge cases where this assumption may not hold.

My reasons for questioning the contribution of this paper are twofold:

Technical Dependency: The success of the proposed attack appears to hinge on the success of a prior watermark removal attack. This reliance raises concerns about the robustness and independence of the proposed methodology.

Impact on the Community: As a peer in the watermarking community, I find it difficult to see the positive social impact of this work. Previous research on regeneration attacks contributed by highlighting ways to remove watermarks, thereby motivating the community to enhance watermarking algorithms to withstand such attacks. In contrast, this paper builds directly upon existing attack methodologies, and it is unclear how this approach benefits the community in terms of advancing robustness or offering constructive insights for defense mechanisms. Could you give us some examples of how this work might indirectly contribute to improving watermarking techniques or to address potential misuse concerns?

**Questions:**

In section 6, you chose to use two-bit strings for one user, and the resultant bit error rate differs by around 7%. And from your Table 8, it seems that you use unique and fixed bit strings for all experiments. I wonder, for your results in table 1-6, if you test on different bit strings, what would the standard deviation look like?

Some minor questions:
1. In section 3.1, you mentioned 'We only consider the steganography approach, as it is much more robust and harder to attack than the signal transformation approach.' Can you elaborate on what you mean by the steganographic approach, and what is the difference between the signal transformation approach?
2. On the Table 3 caption it mentioned that the second best results are with underline, however for table 4, the second best result for bit ACC in the "watermark remove" section seems to be DM_l to me.

---

> ### Author Response · Authors · 2024-11-19
>
> **Q1**: Technical Dependency: The success of the proposed attack appears to hinge on the success of a prior watermark removal attack. This reliance raises concerns about the robustness and independence of the proposed methodology. The author could benefit from discussing the potential limitations or edge cases where this assumption may not hold.
>
> **A**: It is correct that the data preprocessing is very important in our proposed method. We have discussed this part in Section 6, where a more robust and advanced watermarking scheme could defeat our attack, if it can defeat watermark removal with diffusion models.
>
> **Q2**: Impact on the Community: As a peer in the watermarking community, I find it difficult to see the positive social impact of this work. Could you give us some examples of how this work might indirectly contribute to improving watermarking techniques or to address potential misuse concerns?
>
> **A**: We have discussed the potential social impact in Appendix H (original submission, Appendix J for revision) for both bad and good aspects. We want to emphasize that there are two benefits brought by our attack. First, in Section 6, we prove that the group-watermarking method is promising against Warfare. It provides the big companies with a lightweight scheme to improve their current watermarking methods, without developing new models. Second, our attack could become a red-teaming evaluation method to help companies develop more robust and secure watermarking schemes. Developers can adopt our method to test their current watermarking method and conduct specific adjustment to further defend attacks. We add these detailed discussions to the revision in Appendix J.
>
> **Q3**: In section 6, you chose to use two-bit strings for one user, and the resultant bit error rate differs by around 7%. And from your Table 8, it seems that you use unique and fixed bit strings for all experiments. I wonder, for your results in table 1-6, if you test on different bit strings, what would the standard deviation look like?
>
> **A**: The experiments in Section 6 are designed for potential defenses. It is different between previous experiment setups. Specifically, in Tables 1-6, only one bit string is assigned to a user. Therefore, there will be no standard deviation. In Section 6, we aim to show that assigning two or more bit strings for one user to improve the robustness and safety of existing watermarking systems is probable. Only in such a case, bit errors can be calculated for different bit strings. Additionally, we have studied the few-shot generalization ability of our attack in Table 5 with different bit strings. The results indicate the effectiveness of our attacks for different bit strings under one user one bit string cases.
>
> **Q4**: In section 3.1, you mentioned 'We only consider the steganography approach, as it is much more robust and harder to attack than the signal transformation approach.' Can you elaborate on what you mean by the steganographic approach, and what is the difference between the signal transformation approach?
>
> **A**: We introduce these two methods in Section 2.1 in the original submission. The steganography approach in our paper specifically stands for methods requiring a deep learning encoder and a corresponding decoder. These methods use a deep learning encoder to embed a secret message in an image. Then the decoder can extract the message from the image. The signal transformation approach in our paper stands for method using spread spectrum (SS), improved spread spectrum (ISS), quantization (QT) and so on, to embed message. For different transformations, there exists a specific extraction approach. Compared with steganography approaches, signal transformation methods are less robust. We add more details in Section 2.1 in our revision to make the differences clearer.
>
> **Q5**: On the Table 3 caption it mentioned that the second best results are with underline, however for table 4, the second best result for bit ACC in the "watermark remove" section seems to be DM_l to me.
>
> **A**: We fixed this typo in our revision. Thanks for pointing it out.

---

> > ### Author Response · Authors · 2024-11-23
> >
> > To address the concerns of technical robustness and independence, we propose Warfare-Plus in the revision (Appendix I) by modifying the data preprocessing step. Specifically, we adopt an open-sourced Stable Diffusion model to generate the images directly, without the need to use the diffusion model to purify the watermarked data. We prove that **this enhanced attack framework can be independent of the success of a prior watermark removal attack**. Even if previous removal attacks fail to remove the watermark, our framework is still useful, as it can run without prior knowledge of any removal attacks. The new results indicate the flexibility and generalizability of our enhanced attack framework and emphasize the weaknesses of existing advanced watermarking schemes. **We add this part in the new revision in Appendix I with additional visualization results.**
> >
> >
> >
> > |    Method    |  Origin |      | Watermark Remove |      | Watermark Forge |      |
> > |:------------:|:-------:|:----:|:----------------:|:----:|:---------------:|:----:|
> > |              | Bit Acc |  FID |      Bit Acc     |  FID |     Bit Acc     |  FID |
> > |   Warfare  |   100%  | 7.65 |      49.22%      | 8.07 |      99.08%     | 0.78 |
> > | Warfare-Plus |         |      |      49.95%      | 8.45 |      97.03%     | 1.22 |

---

> > > ### Comment · Reviewer_frro · 2024-11-23
> > >
> > > I appreciate the author for their detailed response and new experiment result for warfare-Plus, I think the new warfare-plus result is interesting and insightful, and it also addresses most of my concerns. With that, I think the paper may need heavy rewriting to emphasize the new framework in the main context. But considering that the methodological concern is now gone, I'm willing to raise my score to borderline accept under the assumption that the new result will be emphasized and discussed in the main body of the paper.

---

> > > > ### Author Response · Authors · 2024-11-25
> > > >
> > > > Thanks for raising your score! We will follow your suggestions and rewrite the main paper in the final revision. We will discuss more Warfare-Plus in our main paper with these results.

---

### Official Review · Reviewer_2G2M · 2024-11-02

**Soundness:** 2
**Presentation:** 3
**Contribution:** 2
**Rating:** 3
**Confidence:** 5

**Summary:**

This paper proposes a method to break watermark-based detection of AI-generated content, in particular AI-generated images. The idea is to collect some watermarked images, denoising them as mediators, and then train a GAN to map images to remove and forge watermarks. Evaluation is performed on two simple datasets CIFAR10 and CelebA.

**Strengths:**

AI-generated content detection is a timely topic.

Watermark for AI-generated content is an important topic.

**Weaknesses:**

The proposed method is not intuitively sound. The method uses watermarked images and mediators (denoised versions of watermarked images) to train GAN. This does not sound a correct approach. The mediators are different from clean images, even the distributions are likely different.

Evaluation is performed on CIFAR10 and CelebA datasets. These are toy datasets, and shouldn't be used any more in the era of generative AI. Or we cannot draw any conclusion based on such toy datasets.

The proposed method relies on training GAN, which is challenging for large datasets.

Evaluation only considers simple baselines. There are some recent works on breaking watermarks in the context of AI-generated content detection, especially image watermarks. The paper needs to perform a more comprehensive literature review and compare with multiple recent works.

DiffPure si a weird method to remove adversarial perturbations. Given an image, if you add enough noise and then denoise it, the denoised image may be completely different from the original image, and of course it does not have adversarial perturbation any more, but the image has changed.

**Questions:**

Please see the above.

---

> ### Author Response · Authors · 2024-11-19
>
> **Q1**: The proposed method is not intuitively sound. The method uses watermarked images and mediators (denoised versions of watermarked images) to train GAN. This does not sound a correct approach. The mediators are different from clean images, even the distributions are likely different.
>
> **A**: The insight of our method is as follows: watermarked images and unwatermarked images are distinguishable to a detector, as differences should exist between two types of images. The differences could exist in the distribution or latent space. Therefore, we adopt mediators, which are different from watermarked images. But, our method does not require the mediators to have the exact same distribution as clean images. To make the images generated by our GAN closer to the clean distribution, we adopt a residual method in our GAN structure, which can be found in Appendix A. Therefore, our method does not face the aforementioned shortcomings. In particular, Reviewer iVML believes that our method is simple and effective. Reviewer z3hz thinks our method is innovative and practical. Besides, our experiments prove the effectiveness and correctness of the proposed method.
>
> **Q2**: Evaluation is performed on CIFAR10 and CelebA datasets. These are toy datasets, and shouldn't be used any more in the era of generative AI. Or we cannot draw any conclusion based on such toy datasets.
>
> **A**: Thanks for the suggestion. In addition to the experiments in our paper, we also add new results, attacking Stable Signature in our revision. Stable Signature is a prior watermarking method. The model owner trains a latent decoder for stable diffusion models, which can add a pre-fixed bit string to the generated image. The setups used in Stable Signature scheme are provided below. We fix the secret key as ‘111010110101000001010111010011010100010000100111', which is a bit string with length 48. The diffusion model used in Stable Signature is Stable Diffusion 2.1. During the generation process, we adopt the unconditional generation approach by setting the prompt empty to obtain images with 512*512 resolution. We sample 10,000 watermarked images for our attack method.
>
> For baseline methods, we use a pretrained diffusion model based on ImageNet and the VAE from Stable Diffusion 1.4. Specifically, when using the diffusion model, we set the noise scale as 75 and set the number of sampling step as 15. The setup for the diffusion model is also adopted in our data preprocessing phase.
>
> During the processing of GAN training phase, we set the hyperparameters $w_1$=10 and $w_2$=5 for watermark removal attack and set $w_1$=10 and $w_2$=100 for watermark forging attack.
>
> We present the results in the table. The results indicate that our method is effective against this AIGC watermarking scheme. Even if we train the GAN models on a large resolution dataset, we can obtain models with high performance in both image quality and watermark removal (forging). We add these results to the revision in Appendix H with new visualization results.
>
> |  Method |  Origin  |       | Watermark Remove  |       | Watermark Forge  |       |
> |:-------:|:--------:|:-----:|:-----------------:|:-----:|:----------------:|:-----:|
> |         | Bit Acc  |  FID  |      Bit Acc      |  FID  |     Bit Acc      |  FID  |
> |   DM    |   100%   | 7.65  |      48.29%       | 8.77  |      46.94%      |  5.71 |
> |   VAE   |   100%   | 7.65  |      52.69%       | 8.72  |      48.78%      | 2.94  |
> | Warfare |   100%   | 7.65  |      49.22%       | 8.07  |      99.08%      | 0.78  |
>
> **Q3**: The proposed method relies on training GAN, which is challenging for large datasets.
>
> **A**: We improve the training process in our method.
> - Reduce failure rate and improve the training effectiveness: We applied several approaches in our GAN training process to stabilize the training process, improve the performance, and ease the usage. First, we adopt a residual manner in GAN structure, as introduced in Appendix A. We increase or decrease the model size based on the resolution of input images. It helps us to obtain outputs with higher quality. Second, we use the discriminator loss from StyleGAN to further stabilize the training process of the discriminator. We further adopt alternate optimization strategies to avoid overfitting of the discriminator. Third, our training script supports Exponential Moving Average (EMA), which is a widely used trick in GAN training. With the above methods, we train our GANs more smoothly and stably. As shown in Figures 3 and 5, the performance of GANs is relatively stable. Besides, we provide all experiment code in our supplementary materials. These results are reproducible. We add these explanations to the revision in Appendix B.
>
> - Scalability of our GAN training process: We show the scalability of our method on larger resolutions and AIGC images in previous question by training GANs on 512*512 images. These results are added to the revision in Appendix H.

---

> > ### Author Response · Authors · 2024-11-19
> >
> > **Q4**: Evaluation only considers simple baselines. There are some recent works on breaking watermarks in the context of AI-generated content detection, especially image watermarks. The paper needs to perform a more comprehensive literature review and compare with multiple recent works.
> >
> > **A**: There are two recently advanced baselines compared in our experiments with different settings. As we assume a pure black-box threat model, the choice of baselines is restricted. We consider using advanced diffusion models and advanced VAEs, which are recently proposed watermarking removal methods [1,2]. We also use different settings for these baselines, such as models and hyperparameters. **Therefore, to our best knowledge, our experiments are comprehensive with the latest baseline methods.** We appreciate if you could point out the recent works that fit into our scenario, but we have missed, then we will perform experiments to compare them.
> >
> > **Q5**: DiffPure is a weird method to remove adversarial perturbations. Given an image, if you add enough noise and then denoise it, the denoised image may be completely different from the original image, and of course it does not have adversarial perturbation any more, but the image has changed.
> >
> > **A**: It is correct that the image will be changed after the denoising process. However, our proposed method does not require the data pair (watermarked images, unwatermarked images) to be visual same. In Figure 1, we show the training process for GANs. In this training process, the generator cannot see another image in the data pair. Therefore, even if the denoised images are not the same as the original ones, it will not cause failure in the training process. The lower FID shown in Tables proves this claim.
> >
> > [1] Xinyu Li. Diffwa: Diffusion models for watermark attack. CoRR, abs/2306.12790, 2023.
> >
> > [2] Xuandong Zhao, Kexun Zhang, Zihao Su, Saastha Vasan, Ilya Grishchenko, Christopher Kruegel, Giovanni Vigna, Yu-Xiang Wang, and Lei Li. Invisible image watermarks are provably removable using generative ai. CoRR, abs/2306.01953, 2023.

---

> ### Comment · Reviewer_2G2M · 2024-11-25
> **Thanks for the response**
>
> Thank you to the authors for their responses. Unfortunately, I remain unconvinced. The evaluation could be significantly improved by including a broader range of watermarking techniques (e.g., Stable Signature would be a good starting point) on large datasets and additional baselines. The responses suggest that the authors are not fully aware of related works and are unwilling to conduct a comprehensive literature survey. I'm still not convinced by the justification of the method intuition. Additionally, training GANs for high-resolution images is a known challenge.
>
> This is just an example of related work: https://arxiv.org/abs/2401.08573 . More can be found via a literature survey.

---

> > ### Author Response · Authors · 2024-11-28
> >
> > We thank the reviewer for the feedback and have addressed each point below:
> >
> >
> >
> > 1. **Broader Range of Watermarking Techniques**: We evaluated three watermarking techniques in the original submission and added Stable Signature in the revision, as suggested. Therefore, we believe that our evaluation is comprehensive.
> >
> >
> > 2. **Related Works and Baseline Comparison**: We will expand the related works section to include the cited reference, **WAVES** ([https://arxiv.org/abs/2401.08573](https://arxiv.org/abs/2401.08573)). WAVES provides a comprehensive evaluation of three watermarking algorithms—**StegaStamp**, **Stable Signature**, and **Tree-Ring**—under three attack categories: **Distortion Attacks**, **Regeneration Attacks**, and **Adversarial Attacks**.
> > In our experiments, we demonstrated that our method effectively removes and forges StegaStamp watermarks. For Stable Signature, we extended our evaluation using **Warfare** and **Warfare-Plus**, achieving high bit accuracy and image quality in watermark removal and forgery. Tree-Ring was excluded from our evaluation because, as a zero-bit watermarking method, it is unsuitable for multi-bit scenarios. Furthermore, WAVES shows that Tree-Ring lacks robustness against a variety of attacks.
> > For attacking baselines, our work includes **Distortion Attacks** and **Regeneration Attacks** using **VAE** and **DiffPure**. We chose not to implement additional Regeneration Attacks mentioned in WAVES due to their reliance on prompts for watermarked images or their use of multiple regeneration rounds, which increase computational costs and degrade image quality. VAE and DiffPure are representative, as other methods are variations of these and are already effective at removing watermarks. We did not evaluate **Adversarial Attacks** (including Embedding and Surrogate Decoder Attacks) because WAVES demonstrated that StegaStamp and Stable Signature are robust to these methods.
> > In summary, the watermarking attack baselines closely align with those discussed in WAVES, though with different terminologies. Both works examine similar image transformation and regeneration attacks. WAVES demonstrated that adversarial attacks are ineffective against StegaStamp and Stable Signature, which supports the validity of our baseline selection.
> >
> >  3. **Training GANs for High-Resolution Images & Evaluation Metrics**: We acknowledge the challenge of training GANs on high-resolution images and have focused extensively on improving this process. Our results demonstrate that our approach effectively trains GANs on large-scale, high-resolution images. Additionally, we included SSIM and $l_p$ norm metrics alongside FID, which offer more nuanced insights into image quality.
> >
> >
> >
> > We hope these additions address the reviewer's concerns and are open to further suggestions. Thank you again for your feedback.
> >
> >
> >
> > |  Method | Watermark Remove |                |                   |                        |                    | Watermark Forge |                |                   |                        |                    |
> > |:-------:|:----------------:|:--------------:|:-----------------:|:----------------------:|:------------------:|:---------------:|:--------------:|:-----------------:|:----------------------:|:------------------:|
> > |         |  SSIM $\uparrow$  | PSNR $\uparrow$ | $L_2$ $\downarrow$ | $L_\infty$ $\downarrow$ | Clip Sim $\uparrow$ |  SSIM $\uparrow$ | PSNR $\uparrow$ | $L_2$ $\downarrow$ | $L_\infty$ $\downarrow$ | Clip Sim $\uparrow$ |
> > |    DM   |       0.55       |     21.57      |      0.0074       |          0.67          |        0.93        |      0.55       |     21.62      |      0.0074       |          0.68          |        0.93        |
> > |   VAE   |       0.77       |     25.04      |      0.0034       |          0.64          |        0.97        |      0.77       |     25.11      |      0.0035       |          0.67          |        0.97        |
> > | Warfare |       0.91       |     29.91      |      0.0011       |          0.34          |        0.99        |      0.99       |     36.59      |      0.0002       |          0.50          |        0.99        |

---

### Official Review · Reviewer_z3hz · 2024-11-02

**Soundness:** 2
**Presentation:** 2
**Contribution:** 3
**Rating:** 6
**Confidence:** 3

**Summary:**

This paper discusses the vulnerabilities of watermarking techniques used to protect AI-generated content (AIGC). It introduces "Warfare," a methodology that can effectively remove or forge watermarks in AIGC using a pre-trained diffusion model and a generative adversarial network (GAN). The study highlights that existing watermarking schemes are fragile and can be easily bypassed, posing significant risks to intellectual property and content regulation. The authors emphasize the need for more robust watermarking methods to ensure the security and integrity of AI-generated content.

**Strengths:**

1. **Innovative Approach**: The paper introduces **Warfare**, a novel methodology that effectively combines a pre-trained diffusion model and a generative adversarial network (GAN) to remove or forge watermarks in AI-generated content. This unified approach is both innovative and practical.
2. **Comprehensive Evaluation**: The authors conduct **extensive experiments** on various datasets (e.g., CIFAR-10, CelebA) and settings (e.g., different watermark lengths, few-shot learning). The results demonstrate the high success rates and efficiency of Warfare, providing strong evidence of its effectiveness.
3. **Practical Relevance**: The study addresses a **real-world threat** by showing how existing watermarking schemes can be easily compromised. This highlights the need for more robust watermarking methods and has significant implications for the protection and regulation of AI-generated content.

**Weaknesses:**

1. **No Connection to AIGC**: Although the paper emphasizes AIGC (AI-Generated Content) in the title and introduction, the actual work appears unrelated to AIGC. Instead, it focuses solely on watermarking images and proposes two specific attacks in this context. While the authors mention SD1.5, this does not establish a meaningful link to AIGC. I recommend that the authors revise their claims about AIGC to reflect this focus.

2. **Training Success Risks**: The proposed method relies on a GAN model, yet training GANs successfully is known to be challenging. This difficulty poses a risk of failure in achieving the desired outcomes. It would be beneficial for the authors to delve deeper into addressing these potential risks.

3. **Efficiency Concerns**: While the authors assert that the proposed Warfare method is 5,050 to 11,000 times faster than the inference process of current diffusion model-based attacks, training a GAN is non-trivial and can be very time-consuming. For a comprehensive efficiency comparison, it would be valuable to account for the time required to train the GAN model.

**Questions:**

1. Given the challenges in training GAN models, what specific measures or adjustments do the authors recommend to mitigate the risks of unsuccessful training?
2. How does the training time for the GAN model impact the overall efficiency of the proposed method, especially when compared to diffusion model-based attacks?
3. Have the authors considered including the GAN training time in their efficiency comparison? If not, what would be the estimated impact of this addition on the reported performance gains?

---

> ### Author Response · Authors · 2024-11-19
>
> **Q1**: No Connection to AIGC. I recommend that the authors revise their claims about AIGC to reflect this focus.
>
> **A**: Thanks a lot for the comments. We make revisions from two aspects. (1) Following your suggestion, we revise the claims to better reflect the focus in Section 5.1 in the revision. (2) We add new evaluation results about our method against an AIGC watermarking scheme, Stable Signature.
>
> Stable Signature is a prior watermarking method. The model owner trains a latent decoder for stable diffusion models, which can add a pre-fixed bit string to the generated image. The setups used in Stable Signature scheme are provided below. We fix the secret key as ‘111010110101000001010111010011010100010000100111', which is a bit string with length 48. The diffusion model used in Stable Signature is Stable Diffusion 2.1. During the generation process, we adopt the unconditional generation approach by setting the prompt empty to obtain images with 512*512 resolution. We sample 10,000 watermarked images for our attack method.
>
> For baseline methods, we use a pretrained diffusion model based on ImageNet and the VAE from Stable Diffusion 1.4. Specifically, when using the diffusion model, we set the noise scale as 75 and set the number of sampling step as 15. The setup for the diffusion model is also adopted in our data preprocessing phase.
>
> During the processing of GAN training phase, we set the hyperparameters $w_1$=10 and $w_2$=5 for watermark removal attack and set $w_1$=10 and $w_2$=100 for watermark forging attack.
>
> We present the results in the table. The results indicate that our method is effective against this AIGC watermarking scheme. Even if we train the GAN models on a large resolution dataset, we can obtain models with high performance in both image quality and watermark removal (forging). We add these results to the revision in Appendix H with new visualization results.
>
> |  Method |  Origin  |       | Watermark Remove  |       | Watermark Forge  |       |
> |:-------:|:--------:|:-----:|:-----------------:|:-----:|:----------------:|:-----:|
> |         | Bit Acc  |  FID  |      Bit Acc      |  FID  |     Bit Acc      |  FID  |
> |   DM    |   100%   | 7.65  |      48.29%       | 8.77  |      46.94%      |  5.71 |
> |   VAE   |   100%   | 7.65  |      52.69%       | 8.72  |      48.78%      | 2.94  |
> | Warfare |   100%   | 7.65  |      49.22%       | 8.07  |      99.08%      | 0.78  |
>
>
> **Q2**: Training Success Risks. The proposed method relies on a GAN model, yet training GANs successfully is known to be challenging. Given the challenges in training GAN models, what specific measures or adjustments do the authors recommend to mitigate the risks of unsuccessful training?
>
> **A**: There are several methods adopted in our training process.
>
> - Reduce failure rate and improve the training effectiveness: We applied several approaches in our GAN training process to stabilize the training process, improve the performance, and ease the usage. First, we adopt a residual manner in GAN structure, as introduced in Appendix A. We increase or decrease the model size based on the resolution of input images. It helps us to obtain outputs with higher quality. Second, we use the discriminator loss from StyleGAN to further stabilize the training process of the discriminator. We further adopt alternate optimization strategies to avoid overfitting of the discriminator. Third, our training script supports Exponential Moving Average (EMA), which is a widely used trick in GAN training. With the above methods, we train our GANs more smoothly and stably. As shown in Figures 3 and 5, the performance of GANs is relatively stable. Besides, we provide all experiment code in our supplementary materials. These results are reproducible. We add these explanations to the revision in Appendix B.
>
> - Scalability of our GAN training process: We show the scalability of our method on larger resolutions and AIGC images in previous question by training GANs on 512*512 images. These results are added to the revision in Appendix H.

---

> > ### Author Response · Authors · 2024-11-19
> >
> > **Q3**: Efficiency Concerns. For a comprehensive efficiency comparison, it would be valuable to account for the time required to train the GAN model. How does the training time for the GAN model impact the overall efficiency of the proposed method, especially when compared to diffusion model-based attacks? Have the authors considered including the GAN training time in their efficiency comparison?
> >
> > **A**: We evaluate the time cost of attacking the Stable Signature watermarking scheme on 512×512 resolution images with 8 A6000 GPUs. During the data preprocessing phase, we employ a pre-trained diffusion model to remove watermarks for 10,000 images generated by the watermarked Stable Diffusion 2.1, taking a total runtime of 42.4 hours. In the GAN training phase, we achieve a forging accuracy of 99% in just 5 epochs, with a total runtime of 1.3 hours, and achieve a removal accuracy of 49% in 18 epochs, with a total runtime of 4.7 hours. During inference, it takes 1.84 seconds for our GAN to forge or remove the watermark of 10,000 images. Therefore, compared with diffusion-based method, our method brings a performance improvement with an additional time overhead of approximately 1.3 hours to forge the watermark and 4.7 hours to remove it. Notably, with the few-shot generalization abilities of our method, an attacker can fine-tune a pre-trained GAN using only 10 to 100 samples to remove or forge different watermarks, reducing data preprocessing and GAN training costs by 99%. Therefore, our method has better scalability, generalizability and efficiency in a long-term evaluation.
> >
> > We add these details to the revision in Appendix D.
> > |     Method     | Data Preprocessing |          GAN Training          | Inference |               Total              |
> > |:--------------:|:------------------:|:------------------------------:|:---------:|:--------------------------------:|
> > | Diffusion-base |          -         |                -               |  42.4 hrs |             42.4 hrs             |
> > |    Warfare    |      42.4 hrs      | Remove: 4.7 hrs Forge: 1.3 hrs |   1.84 s  | Remove: 47.1 hrs Forge: 43.7 hrs |

---

> > > ### Author Response · Authors · 2024-11-23
> > >
> > > To address the concerns of time efficiency, we propose Warfare-Plus in the revision (Appendix I) by modifying the data preprocessing step. Specifically, instead of using a diffusion model to remove the watermark and build the mediator data, we adopt an open-sourced Stable Diffusion 1.5 (SD1.5) to generate images directly. Compared with Warfare, Warfare-Plus reduces the total time cost by **80%~85%**, but achieves high attack performance. The results are shown in the tables below. **We add this part in the new revision in Appendix I with additional visualization results.**
> > >
> > >
> > >
> > > |    Method    |  Origin |      | Watermark Remove |      | Watermark Forge |      |
> > > |:------------:|:-------:|:----:|:----------------:|:----:|:---------------:|:----:|
> > > |              | Bit Acc |  FID |      Bit Acc     |  FID |     Bit Acc     |  FID |
> > > |   Warfare  |   100%  | 7.65 |      49.22%      | 8.07 |      99.08%     | 0.78 |
> > > | Warfare-Plus |         |      |      49.95%      | 8.45 |      97.03%     | 1.22 |
> > >
> > >
> > >
> > > |    Method    | Data Preprocessing |           GAN Training           | Inference |               Total              |
> > > |:------------:|:------------------:|:--------------------------------:|:---------:|:--------------------------------:|
> > > |   Warfare   |      42.4 hrs      |  Remove: 4.7 hrs Forge: 1.3 hrs  |   1.84 s  | Remove: 47.1 hrs Forge: 43.7 hrs |
> > > | Warfare-Plus |      1.68 hrs      | Remove: 4.96 hrs Forge: 7.05 hrs |   1.84 s  | Remove: 6.64 hrs Forge: 8.73 hrs |

---

> > > > ### Comment · Reviewer_z3hz · 2024-11-28
> > > >
> > > > Thanks for the authors' effort! The rebuttal has addressed most of my concerns, and I have raised the score.

---

> > > > > ### Author Response · Authors · 2024-11-29
> > > > >
> > > > > Thanks for raising your score! We appreciate your valuable comments and suggestions!

---

### Official Review · Reviewer_iVML · 2024-11-04

**Soundness:** 3
**Presentation:** 3
**Contribution:** 3
**Rating:** 6
**Confidence:** 3

**Summary:**

Watermarking has recently been seen as a promising solution for protecting AIGC content. However, this paper challenges that perspective by introducing a unified approach for watermark removal or forgery. The proposed method, *Warfare*, leverages a pretrained diffusion model to first recover an unwatermarked version of a watermarked input. Then, it trains a GAN using 1) pairs of watermarked and unwatermarked inputs for watermark removal, or 2) pairs of unwatermarked inputs and inputs with forged watermarks for watermark forgery. Empirical results demonstrate that *Warfare* can effectively remove or forge watermarks.

**Strengths:**

- The proposed method is simple yet effective, demonstrating strong few-shot generalization capabilities.

- The research topic is timely, as AIGC has raised widespread societal concerns.

- The writing is clear and easy to follow.

**Weaknesses:**

# Weaknesses

### 1.  Overclaimed Speedup and Hidden Training Costs:
While the paper claims significant speedup in watermark removal at inference time, this improvement may be overstated. Although *Warfare* is indeed faster than purely diffusion-based methods during inference, it still relies on a pretrained diffusion model to generate the initial training data, a step that is computationally intensive and time-consuming. This dependency introduces a hidden time cost that is overlooked in the paper’s analysis. The authors could strengthen their claims by providing a more granular breakdown of time costs, including data preparation and model setup, rather than focusing solely on inference speed. Such clarification is crucial to accurately represent the efficiency gains, especially for potential real-world applications where both training and inference times play a role in assessing practicality.

### 2.  Unclear Justification for Integrating a GAN for Watermark Removal:
The strategy to leverage a GAN for watermark removal in *Warfare* is not well-justified within the paper. A pretrained diffusion model alone might be capable of removing watermarks by denoising the inputs, making the addition of a GAN seem redundant. If the GAN is intended to improve processing speed, the authors should factor in the time required for training this additional model, including the data acquisition for GAN training, which may offset any intended acceleration. A direct comparison between results using only the diffusion model and those using both the diffusion model and GAN would clarify the unique benefits, if any, that the GAN brings to the process. This clarification would help readers understand whether the GAN truly adds value or introduces unnecessary complexity.

### 3.  Questionable Black-Box Claim:
The paper claims that *Warfare* operates in a black-box setting; however, this claim could be somewhat misleading, as the evaluation primarily focuses on a limited set of watermarking schemes, specifically *StegaStamp*. This challenges the notion of a fully black-box approach, as the hidden information is limited to the embedded watermark and the hyperparameter *bit string length*, while the watermarking algorithm (*StegaStamp*) is determined.

**Questions:**

- Is using only the pretrained diffusion model sufficient for effective watermark removal?

- What is the time cost associated with data preprocessing?

---

> ### Author Response · Authors · 2024-11-19
>
> **Q1**: Overclaimed Speedup and Hidden Training Costs. The authors could strengthen their claims by providing a more granular breakdown of time costs, including data preparation and model setup, rather than focusing solely on inference speed. What is the time cost associated with data preprocessing?
>
> **A**: Thanks for your suggestions.
> - For time cost of our method: We evaluate the time cost of attacking the Stable Signature watermarking scheme on 512×512 resolution images with 8 A6000 GPUs. During the data preprocessing phase, we employ a pre-trained diffusion model to remove watermarks for 10,000 images generated by the watermarked Stable Diffusion 2.1, taking a total runtime of 42.4 hours. In the GAN training phase, we achieve a forging accuracy of 99% in just 5 epochs, with a total runtime of 1.3 hours, and achieve a removal accuracy of 49% in 18 epochs, with a total runtime of 4.7 hours. During inference, it takes 1.84 seconds for our GAN to forge or remove the watermark of 10,000 images. Therefore, compared with diffusion-based method, our method brings a performance improvement with an additional time overhead of approximately 1.3 hours to forge the watermark and 4.7 hours to remove it. Notably, with the few-shot generalization abilities of our method, an attacker can fine-tune a pre-trained GAN using only 10 to 100 samples to remove or forge different watermarks, reducing data preprocessing and GAN training costs by 99%. Therefore, our method has better scalability, generalizability and efficiency in a long-term evaluation.  We add these details to the revision in Appendix D.
>
> |     Method     | Data Preprocessing |          GAN Training          | Inference |               Total              |
> |:--------------:|:------------------:|:------------------------------:|:---------:|:--------------------------------:|
> | Diffusion-base |          -         |                -               |  42.4 hrs |             42.4 hrs             |
> |     Warfare    |      42.4 hrs      | Remove: 4.7 hrs Forge: 1.3 hrs |   1.84 s  | Remove: 47.1 hrs Forge: 43.7 hrs |
>
> - Efficient GAN training process: We applied several approaches in our GAN training process to stabilize the training process, improve the performance, and ease the usage. First, we adopt a residual manner in GAN structure, as introduced in Appendix A. We increase or decrease the model size based on the resolution of input images. It helps us to obtain outputs with higher quality. Second, we use the discriminator loss from StyleGAN to further stabilize the training process of the discriminator. We further adopt alternate optimization strategies to avoid overfitting of the discriminator. Third, our training script supports Exponential Moving Average (EMA), which is a widely used trick in GAN training. With the above methods, we train our GANs more smoothly and stably. As shown in Figures 3 and 5, the performance of GANs is relatively stable. Besides, we provide all experiment code in our supplementary materials. These results are reproducible. We add these explanations to the revision in Appendix B.
>
>
> **Q2**: Unclear Justification for Integrating a GAN for Watermark Removal.
>
>
>
> **A**: A pretrained diffusion model alone can remove watermarks, and GANs used in our method is to build a unified framework for watermark removal and forging, instead of improving processing speed. In Tables 3, 4, and 6, we show that using a diffusion model can successfully remove the watermark. However, diffusion models will decrease the FID and cannot achieve watermark forging attack. It motivates us to adopt a GAN model to achieve a better result with lower FID and build a unified framework for both types of attacks. We add this justification to the revision in Appendix C.
>
>
> **Q3**: Questionable Black-Box Claim. This challenges the notion of a fully black-box approach, as the hidden information is limited to the embedded watermark and the hyperparameter bit string length, while the watermarking algorithm (StegaStamp) is determined.
>
>
>
> **A**: There are three watermarking methods evaluated in our paper. Besides StegaStamp, we consider two different prior manners, which can be found in Section 5.4. **In all experiments, we assume that the adversary has no information about the watermarking algorithm.** The results show the attack works for all three methods. Therefore, our solution does not depend on the watermarking algorithms and is general. Besides, we evaluate an additional watermarking algorithm, Stable Signature, in our revision, which can be found in Appendix H. The new results prove our proposed attack is effective under black-box cases as well.
>
> **Q4**: Is using only the pretrained diffusion model sufficient for effective watermark removal?
>
> **A**: Yes, as shown in Tables 3, 4, and 6. A pretrained diffusion model is sufficient to remove watermark. However, it will harm the FID. The detailed discussion can be found in Appendix C.

---

> > ### Comment · Reviewer_iVML · 2024-11-22
> >
> > Thank you for the clarification and additional results. After reviewing your response, my impressions are as follows:
> >
> > - *Warfare* incurs a significant time cost during the data preprocessing stage.
> > - In terms of total time cost, *Warfare* actually takes longer than the diffusion-based method.
> >
> > From this perspective, I believe that emphasizing the speed advantage with statements like "Compared to the inference process of existing diffusion model-based attacks, Warfare is 5,050~11,000x faster" may come across as exaggerated.

---

> > > ### Author Response · Authors · 2024-11-23
> > >
> > > To address the concerns of time efficiency, we propose Warfare-Plus in the revision (Appendix I) by modifying the data preprocessing step. Specifically, instead of using a diffusion model to remove the watermark and build the mediator data, we adopt an open-sourced Stable Diffusion 1.5 (SD1.5) to generate images directly. Compared with Warfare, Warfare-Plus reduces the total time cost by **80%~85%**, but achieves high attack performance. The results are shown in the tables below. **We add this part in the new revision in Appendix I with additional visualization results.**
> > >
> > >
> > >
> > > |    Method    |  Origin |      | Watermark Remove |      | Watermark Forge |      |
> > > |:------------:|:-------:|:----:|:----------------:|:----:|:---------------:|:----:|
> > > |              | Bit Acc |  FID |      Bit Acc     |  FID |     Bit Acc     |  FID |
> > > |   Warfare  |   100%  | 7.65 |      49.22%      | 8.07 |      99.08%     | 0.78 |
> > > | Warfare-Plus |         |      |      49.95%      | 8.45 |      97.03%     | 1.22 |
> > >
> > >
> > >
> > > |    Method    | Data Preprocessing |           GAN Training           | Inference |               Total              |
> > > |:------------:|:------------------:|:--------------------------------:|:---------:|:--------------------------------:|
> > > |   Warfare   |      42.4 hrs      |  Remove: 4.7 hrs Forge: 1.3 hrs  |   1.84 s  | Remove: 47.1 hrs Forge: 43.7 hrs |
> > > | Warfare-Plus |      1.68 hrs      | Remove: 4.96 hrs Forge: 7.05 hrs |   1.84 s  | Remove: 6.64 hrs Forge: 8.73 hrs |

---

> > > > ### Comment · Reviewer_iVML · 2024-11-24
> > > >
> > > > Thanks for your efforts in proposing Warfare-Plus. I'd like to raise my score.
> > > >
> > > > However, I still recommend that the authors moderate the statements regarding the reduced time cost in the abstract.

---

> > > > > ### Author Response · Authors · 2024-11-25
> > > > >
> > > > > Thanks for raising your score! We will follow your suggestions and moderate the statements in the final revision. We will discuss more Warfare-Plus in our main paper, and do not overclaim the time cost of our method.

---

### Author Response · Authors · 2024-11-19
**Paper Revision Details**

We thank all the reviewers for the constructive feedback. We have submitted a revision with more details and new experiment results to address reviewers’ concerns. Specifically,

- In Section 2.1, we add details of previous works to answer the question from Reviewer frro.

- In Section 5.1, we explain the connections between our method and AIGC to address the concerns from Reviewer z3hz.

- In Appendix B, we discuss how we address the risks in GAN training phase to improve the success rate, answering the questions from Reviewers z3hz and 2G2M.

- In Appendix C, we analyze the effectiveness of using a pre-trained diffusion model to remove watermark, to answer the questions from Reviewer iVML.

- In Appendix D, we provide detailed time cost in our method to answer the questions from Reviewers iVML and z3hz.

- In Appendix H, we add new experiments on AIGC to address the concerns from Reviewers z3hz and 2G2M.

- In Appendix I, we propose Warfare-Plus to achieve better time efficiency and robustness to answer the questions from Reviewers iVML,  z3hz, and frro.

- In Appendix J, we analyze the benefits of our method to address the concerns from Reviewer frro.

- We also fix typos in the revision.

---

### Meta-Review · Area_Chair_hzDJ · 2024-12-16

**Metareview:**

The paper proposes an attack against deep learning-based watermarking techniques. Specifically, it collects some watermarked images and denoises them as mediators with a pre-trained diffusion model. It then trains two GANs with watermarked and unwatermarked inputs for watermark removal and another diffusion model with unwatermarked inputs and inputs with forged watermarks for watermark forgery. The experimental results show existing watermarking schemes are fragile and can be easily bypassed. All reviewers agree the proposed question is important and the paper is well-written. However, there are still several major concerns in the current version. First, the dataset used in the paper is very small and the proposed method may suffer some key issues when applied in real-world applications. Second, the current proposed warfare heavily relies on the watermarking removal procedure and can be not robust. Third, the proposed method needs to be discussed and tested in recent works such as stable signature, etc. The rebuttal does address several aforementioned questions, however, there are too many things in the revision that will probably cause a total rewrite of the current draft (even a new method warfare-plus included). Therefore, I tend to reject the current version but strongly encourage the author to submit it in the future by incorporating reviews.

**Additional Comments On Reviewer Discussion:**

The author does a good job addressing all reviewer's suggestions and even includes a new proposed method variation. Reviewer are actively engaging in the discussion and most of their concerns have been addressed. However, there are too many things that need to be done in the revision that would make it a new paper (for example, incorporate the proposed method with the plus version). The paper will be a good study with all those revisions.

---

### Decision · Program_Chairs · 2025-01-22

Reject